# Ageing does not impair motor neuron adaptations: comparable motor unit responses to strength training in young and older adults

Andrea Casolo[1] , Stefanie Del Vecchio[2], Benjamin I. Goodlich[3] , Bastian Schrader[4], Stefano Nuccio[5] , Edoardo Lecce[5] , Ilenia Bazzucchi[5] , Luca Angius[6] , Francesco Felici[5] , Joachim Schrader[4], Dario Farina[7] and Alessandro Del Vecchio[8]

[1]*Department of Biomedical Sciences, University of Padova, Padua, Italy*
[2]*Klinikum Nürnberg, Nürnberg, Germany*
[3]*School of Health, University of the Sunshine Coast, Sunshine Coast, Queensland, Australia*
[4]*Department of Cardiology, University of Oldenburg, Klinikum, Oldenburg, Germany*
[5]*Department of Movement, Human and Health Sciences, University of Rome 'Foro Italico', Rome, Italy*
[6]*Department of Sport, Exercise, and Rehabilitation, Faculty of Health and Life Sciences, Northumbria University, Newcastle upon Tyne, UK*
[7]*Department of Bioengineering, Imperial College London, London, UK*
[8]*Department of Artificial Intelligence in Biomedical Engineering, Friedrich-Alexander University, Erlangen-Nürnberg, Germany*

Handling Editors: Karyn Hamilton & Kevin Murach

The peer review history is available in the Supporting Information section of this article (https://doi.org/10.1113/JP290541#support-information-section).

**Abstract figure legend** Ageing is associated with reduced motor neuron excitability and lower motor unit (MU) discharge rates, contributing to impaired voluntary force production. Using high-density surface EMG and longitudinal

motor unit tracking, we show that 4 weeks of isometric strength training increases discharge rates and estimates of persistent inward currents (PICs) in older adults, reflecting enhanced intrinsic motor neuron excitability and synaptic input. These neural adaptations were associated with a 17.6% increase in maximal voluntary force, demonstrating that the ageing neuromuscular system retains a robust capacity for functional adaptation in response to targeted strength training.

**Abstract** Ageing induces structural and functional changes in the neuromuscular systems that impair voluntary force production, compromising daily function and wellbeing. We examined whether older adults preserve the capacity for motor unit adaptations to a short-term strength training intervention previously shown to enhance neural drive to muscle in young adults. Twenty–three older adults were assigned to a training group (INT, $n = 13$, $71 \pm 4$ years of age) or a control group (CON, $n = 10$, $69 \pm 2$ years of age) and completed pre- and postintervention assessments of ankle dorsiflexor maximal voluntary force (MVF). Motor unit behaviour was analysed from high–density surface EMG recorded from tibialis anterior during submaximal trapezoidal contractions. The INT group performed a 4 week supervised isometric strength training programme, whereas the CON group maintained habitual activity. High–density surface EMG signals were decomposed into individual motor units, tracked longitudinally across sessions. Training increased MVF by 17.6% and enhanced motor unit discharge rate at recruitment ($+8.2\%$, $P = 0.031$) and constant force ($+11.3\%$, $P < 0.001$), without changes in recruitment or derecruitment thresholds. Estimates of persistent inward currents (delta frequency) increased ($+1.0$ pulses per second) and were positively correlated with changes in discharge rate, which, in turn, were correlated with gains in MVF ($r_{\mathrm{rm}} = 0.54$–$0.57$, $P < 0.05$). This pattern suggests that enhanced intrinsic excitability and synaptic input to motor neurons contributed to improvements in strength. These results demonstrate that, despite age-related motor unit remodelling, the ageing nervous system remains responsive to targeted strength training, preserving the capacity for meaningful neural adaptations.

(Received 13 November 2025; accepted after revision 10 February 2026; first published online 14 March 2026)

**Corresponding author** A. Del Vecchio: Department of Artificial Intelligence in Biomedical Engineering, Friedrich-Alexander University Erlangen-Nürnberg, Nürnberger Straße 74, 91052 Erlangen, Germany. Email: alessandro.del.vecchio@fau.de

## Key points

- We assessed whether a short-term intensive strength training intervention, previously shown to increase spinal motor output to the muscle significantly in young adults, would also be effective in older adults.
- High-density surface EMG was used to identify and longitudinally track the same motor units before and after a 4 week isometric strength training intervention.
- We found significant strength gains in older adults, with the increase in muscle force output being positively associated with higher motor unit discharge rate and persistent inward currents, indicating that neural drive enhancement was a key contributor to the observed improvements in force.
- Despite age-related motor neuron remodelling, the older nervous system remains highly responsive to strength training, exhibiting qualitatively similar but attenuated motor unit adaptations compared with young adults.

## Introduction

The age-related decline in maximal voluntary muscular strength and power (Frontera et al., 1991; Skelton et al., 1994; Van Roie et al., 2018) impairs functional capacity, increases fall risk and compromises the performance of daily activities (Bento et al., 2010; Hayashida et al., 2014; Rantanen et al., 1994; Tomás et al., 2018). Although traditionally attributed to reductions in muscle mass and alterations in fibre composition (Evans, 1995),

strength loss often exceeds the degree of muscle atrophy (Delmonico et al., 2009), indicating neural impairments in age-related motor decline (Clark, 2023; Manini et al., 2013). Although neural adaptations to strength training have been documented at both central and peripheral levels (Christie & Kamen, 2010; Kamen & Knight, 2004; Orssatto et al., 2023; Walker, 2021), the extent to which motor neurons adapt, particularly in older adults, and whether these responses differ from those in younger individuals remain unclear.

The motor unit, as the final common pathway for force generation, undergoes several ageing-related morphological and functional alterations (Heckman & Enoka, 2004; Hunter et al., 2016; Liddell & Sherrington, 1925), including reduced excitatory input (McGinley et al., 2010; Opie et al., 2020), diminished intrinsic excitability (Hassan et al., 2021; Orssatto et al., 2021a) and structural degeneration (Hepple & Rice, 2016), leading to motor unit remodelling (Piasecki et al., 2016). These changes impair the ability of motor neurons to sustain high-frequency discharges. As a result, motor unit discharge rate, a key determinant of force modulation and rapid force generation, is typically reduced in older adults (Connelly et al., 1999; Klass et al., 2008; Orssatto et al., 2022).

Strength training is effective in counteracting age-related declines in muscle strength, especially through early neural adaptations (Aagaard et al., 2010; Duchateau et al., 2006; Macaluso & De Vito, 2004; Škarabot et al., 2021). In older adults, indirect evidence includes increased surface EMG amplitude after training (Häkkinen et al., 1998; Moritani & DeVries, 1980), interpreted as enhanced agonist activation. This response is likely to be supported by greater descending drive, increased motor neuron excitability and reduced spinal inhibition, mechanisms that collectively promote greater motor unit recruitment and discharge rates (Lecce et al., 2026; Walker, 2021).

Despite extensive indirect evidence and emerging motor unit-level data from interventions combining resistance training with nutritional supplementation (Nishikawa et al., 2025; Watanabe et al., 2020), direct assessments of motor unit adaptations to strength training alone in older adults remain limited. Only two studies have reported increased discharge rates during maximal (but not submaximal) efforts and reduced after-hyperpolarization duration after short-term resistance training (Christie & Kamen, 2010; Kamen & Knight, 2004), suggesting preserved motor unit adaptations. More recently, Orssatto et al. (2023) reported increased estimates of persistent inward currents (PICs) after 6 weeks of training, indicative of enhanced intrinsic motor neuron excitability. Notably, PIC amplitude gains were moderately correlated with peak discharge rate changes, supporting their role in mediating initial strength gains. Collectively, these findings indicate that, although constrained by age-related changes, motor neurons in older adults retain a meaningful capacity for positive and functionally relevant adaptation.

In young adults, a 4 week isometric strength training intervention has been shown to increase maximal force output primarily through enhanced discharge rates and reduced recruitment thresholds (Del Vecchio et al., 2019), with these modifications likely to be supported by enhanced PIC amplitude and a greater proportion of shared synaptic input (Lecce et al., 2025). Whether older adults exhibit a comparable pattern of adaptation in identical training conditions remains unknown. Recent evidence suggests that young and older adults might exhibit different neural responses even after a single bout of resistance exercise (Nishikawa et al., 2024), supporting the notion that age-related differences in acute motor unit behaviour might influence long-term adaptations. Nonetheless, previous findings of increased discharge rate and PIC estimates following short-term training interventions (Kamen & Knight, 2004; Orssatto et al., 2023) suggest that core mechanisms of motor neuron adaptations are preserved, at least in part, with ageing. Accordingly, we hypothesized that, despite being slower and less excitable, aged motor neurons would maintain the capacity for qualitatively comparable adaptation patterns to those observed in young adults.

In this study, we investigated motor unit adaptations underlying strength gains following a 4 week isometric strength training programme in older adults, applying the same experimental model used by Del Vecchio et al. (2019). The primary aim was to assess changes in motor unit discharge rate, recruitment threshold and intrinsic excitability in response to short-term training. Furthermore, by replicating the protocol and analytical

**Andrea Casolo** is an Assistant Professor at the Department of Biomedical Sciences, University of Padua, Italy. He received a BSc in Sports and Physical Education from the Catholic University of Milan in 2014, an MSc in Health and Physical Activity in 2016, and a PhD in Human Movement and Sport Sciences from the University of Rome 'Foro Italico' in 2020. He subsequently worked as a postdoctoral researcher at Imperial College London, gaining expertise in non-invasive techniques to record motor unit activity *in vivo*. His research focuses on neural control of movement and neuromuscular adaptations to exercise and ageing.

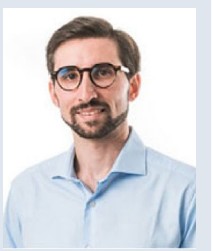

approach previously used in young adults, we have compared the pattern and magnitude of adaptations across age groups using high-density surface EMG (HDsEMG) and longitudinal motor unit tracking. We hypothesized that, although motor neurons in older adults are slower and less excitable, they would still retain the capacity to adapt and display a qualitatively similar pattern of changes to that observed in young adults.

## Methods

### Ethical approval

The study was approved by the local Ethical Committee of the University Medical Centre Göttingen (protocol no. 27-6-19) and adhered to the principles of the *Declaration of Helsinki*, except for registration in a database. All participants provided written informed consent after being fully informed about the experimental procedures and potential risks.

### Participants

A total of 26 healthy, community-dwelling older adults volunteered to take part in this longitudinal interventional study. General inclusion criteria required participants to: (i) be aged $\geq 65$ years; (ii) be free of any lower-limb musculoskeletal disorders, cardiovascular and neurological diseases; (iii) not be taking medications that could influence the monoaminergic system (e.g. selective serotonin or noradrenaline reuptake inhibitors, $\beta$-blockers); and (iv) be untrained, with no prior history of strength or resistance training.

After a familiarization session, participants were randomly assigned to either an intervention group (INT) or a control group (CON) using block randomization to ensure balanced group sizes (Kang et al., 2008). Three participants from the CON group were excluded for not completing the postintervention neuromuscular assessment. Consequently, 23 participants (INT, $n = 13$; CON, $n = 10$) completed all procedures and were included in the final analysis (Table 1).

### Overview

We replicated the experimental protocol and isometric strength training intervention adopted in previous studies (Casolo et al., 2020; Del Vecchio et al., 2019), which addressed the same research question, that is the quantification of changes in discharge characteristics of the same motor units in the tibialis anterior muscle in a cohort of healthy young adults. Accordingly, procedures are only briefly summarized here.

**Table 1. Baseline characteristics of participants by group**

| Variable | Group | | *P*-value |
| | INT | CON | |
|---|---|---|---|
| Age (years) | 71.2 ± 4.7 | 69.0 ± 2.7 | 0.209 |
| Female sex [*n* (% of sample)] | 4 (30.8) | 4 (40.0) | |
| Height (m) | 1.76 ± 0.07 | 1.73 ± 0.09 | 0.297 |
| Body mass (kg) | 84.3 ± 14.9 | 85.7 ± 27.7 | 0.878 |
| MVF (N) | 258.4 ± 75.4 | 280.1 ± 94.4 | 0.548 |

Statistical comparisons were performed with Student's unpaired *t* tests. Data are presented as the mean ± SD. INT, $n = 13$; CON, $n = 10$. Abbreviations: CON, control group; INT, intervention group; MVF, maximal voluntary force.

Participants initially completed a familiarization session (visit 1), during which they were habituated to the experimental set-up and practised performing maximal, rapid and submaximal trapezoidal isometric contractions with the ankle dorsiflexors. Approximately 1 week later, they underwent the baseline neuromuscular assessment (visit 2), which involved simultaneous recording of ankle dorsiflexion force via isometric dynamometry and HDsEMG from the dominant tibialis anterior muscle during both maximal voluntary contractions and submaximal force-matching trapezoidal contractions.

Following the baseline assessment, participants in the INT group undertook a 4 week supervised unilateral isometric strength training programme for the ankle dorsiflexors, consisting of a combination of rapid and sustained contractions. In contrast, participants in the CON group were instructed to maintain their habitual levels of physical activity and dietary habits. Approximately 1 week after the completion of the intervention, all participants underwent the postintervention neuromuscular assessment (visit 3).

### Experimental procedures

**Baseline and postintervention neuromuscular assessment.** Neuromuscular assessments were conducted using a rigid ankle ergometer (v.1.0; OT Bioelettronica, Turin, Italy) equipped with a load cell and secured to a massage table (Fig. 1*A*).

Following a standardized and progressive warm-up, consisting of eight isometric ankle dorsiflexion efforts at self-perceived intensities of maximal voluntary force (MVF; $4 \times 50\%$, $3 \times 70\%$ and $1 \times 90\%$, with 15–30 s rest), participants performed three or four maximal voluntary isometric contractions with ankle dorsiflexors of their dominant leg. During each contraction, participants were

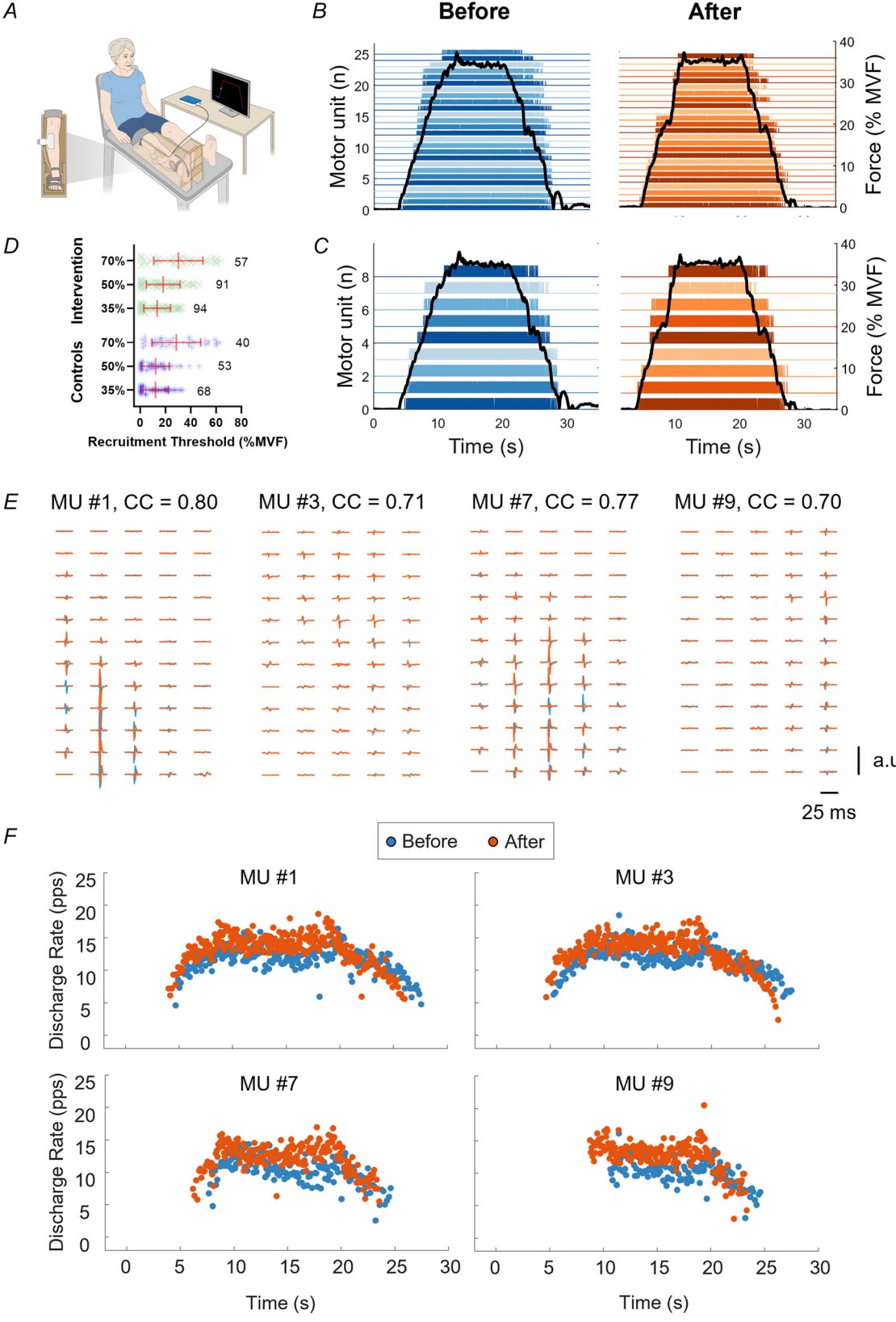

**Figure 1. Overview of the experimental set-up and procedures for data collection and analysis**
*A*, high-density surface EMG signals were recorded from the tibialis anterior muscle during isometric trapezoidal contractions at 35%, 50% and 70% of maximal voluntary force (MVF). *B* and *C*, the recorded signals were

decomposed using a blind-source separation algorithm to identify the discharge timings of individual motor units (*B*), which were then tracked longitudinally across the two recording sessions separated by a 4 week period (*C*). *D* shows a complete overview of the distribution and number of tracked motor units for both groups and contraction levels. In this representative participant from the intervention group, 26 (blue) and 25 (orange) motor units were identified during baseline and postintervention recordings, respectively, for a 35% MVF contraction (black line shows the force trajectory). Among these, nine motor units were tracked successfully across sessions. *E*, multichannel action potential profiles from baseline (blue) and after-training (orange) for four representative motor units (MU #1, #3, #7 and #9) are shown to confirm the similarity of their motor unit action potential shapes. The high two-dimensional cross-correlation coefficients (CC) are reported for each of the four tracked motor units. *F* shows the instantaneous discharge rates for the same four motor units during the trapezoidal contraction. Note that for each motor unit, the instantaneous discharge rate was higher after the strength training intervention.

instructed to 'pull as hard as possible' for 3–5 s while receiving strong verbal encouragement to surpass the peak force recorded in the previous trial, which was displayed on a monitor with a horizontal cursor. Each contraction was separated by ∼30 s of rest.

The highest instantaneous force (i.e. the peak force reached at any single time point) achieved across the three or four maximal efforts was defined as the participant's maximal voluntary isometric force (MVF) of the ankle dorsiflexors and was used as the reference value to determine the submaximal force targets for the trapezoidal contractions. To ensure accurate MVF assessment, participants practised these maximal tasks during the familiarization session under the supervision of an experienced investigator.

Approximately 5 min after completing the maximal isometric contractions, participants performed six trapezoidal contractions: two trials each at 35%, 50% and 70% of MVF, in randomized order. Each contraction consisted of a linear ascending phase from rest to the target force at a constant rate of 5% MVF/s, a 10 s plateau at the target force, and a linear descending phase back to the rest at the same rate. Trials were separated by 3–5 min of rest. Participants were instructed to match a visual force template displayed on a monitor 'as precisely as possible', with real-time visual feedback provided by overlying the live force output onto the reference template.

Participants were instructed to refrain from engaging in strenuous physical activity for 48 h and to avoid caffeine consumption for 24 h prior to each neuromuscular assessment session.

**Training protocol.** The training protocol used in the present study was identical to that used in our previous work (Del Vecchio et al., 2019) and consisted of a combination of rapid and sustained isometric contractions. This unilateral strength training regimen has previously been shown to induce significant neural adaptations at the motor unit level in healthy young adults. The intervention consisted of 12 supervised sessions conducted over 4 weeks (three sessions per week). Each session lasted ∼30 min, with ≥48 h between sessions. All training sessions were performed in a controlled laboratory environment using the same ankle ergometer

(v.1.0; OT Bioelettronica, Turin, Italy) used for the neuromuscular assessments.

Following a standardized warm-up (2 × 50%, 2 × 70% and 1 × 90% of perceived MVF), participants performed three maximal isometric contractions and a series of rapid and sustained isometric ankle dorsiflexion contractions with their dominant foot. For the rapid contractions (four sets of 10 repetitions; 1 min rest between sets and 5 s between repetitions), participants were instructed to contract 'as fast and as hard as possible' to exceed a horizontal cursor set at 75% of MVF within ∼1 s. Contractions were to be performed without any countermovement or pre-tension, and participants were instructed to release the force immediately after reaching the target. For the sustained contractions (three sets of 10 repetitions; 2 min of rest between sets and 2 s between repetitions), participants were asked to match 'as precisely as possible' a visual force template. The template featured a linear increase in force at a constant rate (37.5% MVF/s) up to a target of 75% MVF, which participants were required to maintain for 3 s before returning to rest.

**Force recording.** During neuromuscular assessments and training sessions, participants were comfortably seated on a massage table, with their back against the seatback (Fig. 1*A*). The hip was positioned at ∼120° of flexion (180° = anatomical position), the dominant knee was fully extended (∼180°), and the ankle was positioned at ∼100° of plantar flexion (90° = perpendicular to the tibia). Interindividual differences in lower-limb length were accommodated by adjusting the position of the dynamometer on the table using two adjustable straps during the familiarization session; the same configuration was replicated precisely during all subsequent testing and training sessions. To minimize compensatory movements, the ankle and foot of the dominant leg were securely fastened to the adjustable footplate of the dynamometer using Velcro straps, while the contralateral (non-dominant) leg rested passively on the table.

Ankle dorsiflexion force was measured using a calibrated load cell (CCT Transducer S.A.S., Turin, Italy) connected in series and positioned perpendicularly to an adjustable footplate. The analog force signal was amplified (×500), sampled at 2000 Hz, and digitized via an external

16-bit analog-to-digital converter (Sessantaquattro, OT Bioelettronica, Turin, Italy). Force and HDsEMG signals were acquired and synchronized at the source using OT Biolab software (v.2.0.6352.0, OT Bioelettronica, Turin, Italy). Real-time visual feedback of the force trace and target templates was displayed through the same software.

**HDsEMG recording.** Myoelectrical activity from the tibialis anterior muscle was recorded during submaximal isometric contractions using a high-density adhesive grid consisting of 64 equally spaced electrodes arranged in 13 rows and five columns, with an interelectrode distance of 8 mm (GR08MM1305, OT Bioelettronica, Turin, Italy). Grid placement and orientation followed the procedures established in previous studies (Casolo et al., 2020; Del Vecchio et al., 2019).

Briefly, the muscle belly was identified through palpation, and its borders were outlined with a surgical marker. A 16-electrode dry array was then used to locate the main innervation zone, which is the point along the array where the propagation direction of action potentials reversed both proximally and distally. The identified skin area overlying the muscle was then shaved, lightly abraded, and cleansed with alcohol wipes. The electrode grid was positioned while the participant's dominant leg was extended (∼180°) and the ankle held at ∼100° of plantar flexion, replicating the testing configuration (Fig. 1*A*). The grid was attached to the skin using a disposable bi-adhesive foam layer, and conductive paste (SpesMedica, Battipaglia, Italy) was applied to the cavities near the electrodes to optimize skin-to-electrode contact. Reference electrodes were placed on the ulnar styloid process (main ground) and the medial malleolus (grid reference) of the dominant side. To ensure consistent electrode placement across assessments, the grid profile was marked on the skin with a surgical marker after each recording and training session. HDsEMG signals were recorded in monopolar configuration, band-pass filtered (10–500 Hz, 3 dB bandwidth), and digitized at a sampling rate of 2000 Hz using a 16-bit analog-to-digital converter (Sessantaquattro, OT Bioelettronica, Turin, Italy).

### Data analysis

**Force signal.** The raw voltage signal from the load cell was converted to force (in newtons) and corrected for gravitational offset. The signal was then low-pass filtered using a fourth-order, zero-lag Butterworth filter with a cut-off frequency of 15 Hz.

For each participant, only one of the two isometric trapezoidal contractions per force target (35%, 50% and 70% of MVF) was retained for analysis. The selected trial was the one showing the smallest deviation from the pre-scribed visual force template (Casolo et al., 2021; Nuccio et al., 2020).

**HDsEMG signal.** Monopolar HDsEMG signals recorded during trapezoidal contractions were first band-pass filtered using a second-order, zero-lag Butterworth filter (cut-off frequencies of 20–500 Hz). The filtered signals were then decomposed into individual motor unit spike trains using the validated Convolution Kernel Compensation algorithm implemented in the software DEMUSE (v.5.01; The University of Maribor, Slovenia) (Holobar & Zazula, 2007; Holobar et al., 2014). The automatically identified spike trains were converted into binary format and subsequently inspected and manually edited by an experienced investigator to correct false positives and negatives, following established guidelines (Del Vecchio et al., 2020; Hug et al., 2021). Only motor units exhibiting a reliable discharge pattern were retained for further analysis. Inclusion criteria were a pulse-to-noise ratio of >28 dB (indicating sensitivity of motor unit firing identification of >85%), the absence of physiologically implausible interspike intervals (e.g. <20 ms or >250 ms), and a consistent discharge pattern without excessive irregularity (e.g. coefficient of variation of interspike intervals of >0.4) (Holobar et al., 2014; Škarabot et al., 2023). Potential duplicates, defined as motor units sharing ≥30% of identical discharge instants, were identified using a temporal matching window of 0.5 ms. In such cases, the motor unit with the lower pulse-to-noise ratio was discarded from the dataset. An example of a force signal and the corresponding spike trains of all identified motor units during a representative contraction at 35% MVF are shown in Fig. 1*B*, while Fig. 1*C* illustrates the activity of the pool of motor units successfully tracked across sessions.

For each identified motor unit, basic properties were extracted. Specifically, normalized recruitment and derecruitment thresholds, in addition to average discharge rates across the full contraction and its three distinct phases, were calculated. Recruitment and derecruitment thresholds corresponded to the percentage of force (% MVF) at which the first and last motor unit action potentials were discharged, respectively. Discharge rate was computed across the entire contraction and separately for: (i) the recruitment phase (average of the first four discharges); (ii) the plateau phase (average of the first 10 discharges); and (iii) the derecruitment phase (average of the last four discharges) (Angius et al., 2024; Lecce et al., 2026).

To characterize the overall motor unit population, data from contractions at each force level (35%, 50% and 70% of MVF) were combined for each participant after duplicate removal (Del Vecchio et al., 2020; Martinez-Valdes et al., 2017). To ensure reliable comparison between

motor units across baseline and postintervention assessment, we used a validated tracking approach based on two-dimensional cross-correlation of motor unit action potential waveforms (Martinez-Valdes et al., 2017). This procedure is known to be robust to both physiological and pharmacological manipulations of motor unit discharge characteristics (Del Vecchio & Farina, 2020; Goodlich et al., 2023), ensuring reliable assessment of training-induced changes. Motor unit action potential waveforms were extracted by spike-triggered averaging over a 25 ms time window. A correlation threshold of $\geq 0.70$ was applied to account for slight discrepancies in grid positioning between sessions (Borzuola et al., 2023; Del Vecchio et al., 2019; Orssatto et al., 2023). All motor unit matches were verified visually by an expert operator, as illustrated in Fig. 1 *E* and *F*, which show four representative motor units and their corresponding instantaneous discharge rates. The distribution and number of tracked motor units as a function of recruitment threshold, contraction level and group are shown in Fig. 1*D*.

**Estimation of PICs.** To estimate the contribution of PICs to motor neuron firing, we analysed motor unit onset–offset hysteresis using delta frequency ($\Delta F$) calculation, based on the paired motor unit technique (Gorassini et al., 1998, 2002). $\Delta F$ is defined as the difference in the smoothed discharge rate of a lower-threshold control unit between the recruitment and derecruitment of a higher-threshold test unit (Gorassini et al., 2002). In the present study, $\Delta F$ values are presented as unit-wise averages for all suitable test–control unit pairs, reducing the $\Delta F$ values to one per test unit (Hassan et al., 2021). When a test unit was paired with multiple control units, the resulting $\Delta F$ values were averaged (Hassan et al., 2021; Mesquita et al., 2022; Trajano et al., 2020). Inclusion criteria for valid motor unit pairs were as follows: (i) the test units was recruited $\geq 1$ s after the control unit to ensure full PIC activation; (ii) test and control units had rate–rate correlations of $r^2 > 0.7$ to confirm common synaptic input; and (iii) the control unit showed discharge rate modulation of $\geq 0.5$ pulses per second (pps) while the test unit was active (Jenz et al., 2023). Consistent with previous studies using trapezoidal or triangular contractions of $\leq 40\%$ of MVF for $\Delta F$ estimation (Afsharipour et al., 2020; Orssatto et al., 2021b; Wilson et al., 2015), $\Delta F$ values were only calculated for contractions at 35% MVF. Indeed, accurate $\Delta F$ calculation relies on low-threshold motor units, which are difficult to decompose at higher contraction intensities owing to interference from larger, higher-threshold units in the HDsEMG signal. Accordingly, $\Delta F$ analyses at 50% and 70% MVF were not performed in the present investigation.

## Statistical analysis

Normality of the distribution of data was tested with the Shapiro–Wilk test. When required, the assumption of sphericity was tested using Mauchly's test; if violated, the Greenhouse–Geisser correction was applied.

Baseline between-group differences in anthropometric characteristics, ankle dorsiflexor MVF and the average number of identified motor units (both unmatched and tracked) were assessed using multiple Student's unpaired *t* tests.

The effects of the strength training intervention on MVF were assessed using a two-way repeated-measures ANOVA (time: before *vs.* after; group: intervention *vs.* control). When a significant time × group effect was observed, Bonferroni-corrected *post hoc* comparisons were performed.

Separate generalized linear mixed models (GLMM) were used to evaluate the effects of the intervention on motor unit properties (recruitment/derecruitment threshold, average discharge rate and discharge rate at recruitment, at plateau and at derecruitment), with group and time as fixed effects and with participant as a random intercept [e.g. average discharge rate ∼ group × time + (1|participant ID)]. Significant main effects were followed by Bonferroni-corrected *post hoc* comparisons. Linear mixed models (LMMs) were used to assess changes in $\Delta F$. In the presence of a significant group × time interaction, *post hoc* comparisons were conducted to examine before–after differences within each group with Bonferroni correction applied.

Repeated-measures correlation was used to assess within-participant associations between changes in motor unit discharge rate and changes in MVF across the intervention. The same approach was used to test the association between changes in $\Delta F$ values and changes in motor unit discharge rate. A fixed slope was estimated to derive a single correlation coefficient across participants. Correlation magnitudes were interpreted as follows: $r < 0.1$ = trivial, 0.1–0.3 = small, 0.3–0.5 = moderate, 0.5–0.7 = large, 0.7–0.9 = very large, and $> 0.9$ = nearly perfect (Orssatto et al., 2023).

To compare the magnitude of training-induced adaptations between young and older adults, we compared the present dataset of older adults with data from a previous study in young adults who underwent an identical experimental protocol (Del Vecchio et al., 2019). Separate GLMMs were fitted for each motor unit property, with fixed effects for age (young *vs.* older adults), time (before *vs.* after training) and their interaction, and with a random intercept for participant to account for repeated measures. When a significant age × time interaction was detected, Bonferroni-adjusted *post hoc* comparisons were conducted to compare within-group and between-group differences.

The GLMMs and LMMs were performed using the GAMLj module in Jamovi (v.2.3.28; The Jamovi Project, Sydney, NSW, Australia). Repeated-measures correlation was computed using the *rmcorr* package (Bakdash & Marusich, 2017). Statistical significance was set at $\alpha < 0.05$. Results are reported as estimated marginal means (EMM) and 95% confidence intervals (CI) for linear models and as the mean $\pm$ SD for ANOVA-based analyses. Partial eta squared ($\eta^2_p$) and Cohen's $d$ effect sizes are reported to quantify the magnitude of main and interaction effects.

## Results

### Baseline assessment of older adults

At baseline, no significant differences were observed between the control and intervention groups in age, height, body mass and MVF of the ankle dorsiflexors (Table 1). Likewise, baseline comparisons revealed no significant between-group differences in any motor unit property, including normalized recruitment and derecruitment thresholds, average discharge rate and discharge rate at the recruitment, plateau and derecruitment phases (all $P > 0.05$).

### Motor unit decomposition and tracking

A total of 2672 unique motor units were identified from the tibialis anterior muscle across all participants, contraction intensities and experimental conditions (INT, $n = 1605$; CON, $n = 1067$). The average number of identified motor units per participant and contraction did not differ between the groups, either pre-intervention (INT, $20.7 \pm 6.7$; CON, $17.5 \pm 7.1$; $P = 0.058$) or postintervention (INT, $20.4 \pm 5.3$; CON, $18.1 \pm 6.9$; $P = 0.109$).

A total of 403 motor units (15.1% of the total pool) were tracked successfully between the baseline and post-intervention session (INT, $n = 242$; CON, $n = 161$). The average number of tracked motor units per participant and contraction was comparable between groups (INT, $6.4 \pm 2.4$; CON, $5.4 \pm 2.7$; $P = 0.111$). The average pulse-to-noise ratio of the tracked motor units was $38.7 \pm 5.7$ dB for the INT group and $37.5 \pm 5.8$ dB for the CON group. The average two-dimensional cross-correlation coefficient for the tracked motor units was $0.78 \pm 0.05$ for the INT group and $0.76 \pm 0.05$ for the CON group, supporting the reliability of the tracking procedure.

### Strength training-associated changes in muscle strength and motor unit properties

After 4 weeks of strength training, the MVF of the ankle dorsiflexors increased significantly in the INT group (group $\times$ time interaction, $F_{1,21} = 9.93$, $P = 0.005$, $\eta^2_p = 0.32$), from $258.4 \pm 75.4$ to $305.3 \pm 91.6$ N ($+17.6 \pm 12.1\%$, $P = 0.003$, $d = 1.00$). In contrast, no significant changes were observed in the CON group (pre-intervention, $280.1 \pm 94.4$ N; postintervention, $264.7 \pm 90.6$ N; $P = 0.626$, $d = -0.33$).

Scatterplots of average motor unit discharge rates before and after training revealed a clear upward shift in the INT group, with most data points falling above the line of identity (Fig. 2*A*), indicating increased discharge rates in the same units tracked over time. In contrast, the CON group showed a more balanced distribution around the line of identity, suggesting no systematic change (Fig. 2*B*). This visual trend was confirmed by GLMM, which revealed a significant group $\times$ time interaction [$\chi^2(1) = 11.24$, $P < 0.001$, $\eta^2_p = 0.36$], indicating a differential training effect between groups. *Post hoc* comparisons showed a significant increase in discharge rate in the INT group following training, with EMM increasing from 12.8 pps [95% CI: 11.8, 13.9] pre-intervention to 14.1 pps [13.1, 15.2] post-intervention ($\Delta$EMM = +1.28 pps, $P < 0.001$, $d = 0.68$), corresponding to a $\sim$10.2% increase relative to baseline. In contrast, no significant change was observed in the CON group (pre-intervention, 13.7 pps [12.5, 14.8], post-intervention, 13.6 pps [12.5, 14.8]; $\Delta$EMM = $-0.1$ pps; $P = 1.000$, $d = -0.05$).

When examining specific phases of the contraction, significant group $\times$ time interactions were found for discharge rate at recruitment [$\chi^2(1) = 4.97$, $P = 0.026$, $\eta^2_p = 0.18$; Fig. 2*C*], during the plateau phase [$\chi^2(1) = 12.13$, $P < 0.001$, $\eta^2_p = 0.36$; Fig. 2*D*] and at derecruitment [$\chi^2(1) = 6.10$, $P = 0.014$, $\eta^2_p = 0.22$; Fig. 2*E*]. In the INT group, discharge rate at recruitment increased from 7.73 pps [95% CI: 6.81, 8.65] pre-intervention to 8.36 pps [7.43, 9.29] post-intervention ($\Delta$EMM = +0.63 pps; $P = 0.031$, $d = 0.37$; $\sim$8.2% increase relative to baseline). During the plateau phase, discharge rate increased from 14.2 pps [13.0, 15.4] to 15.8 pps [14.5, 17.0] ($\Delta$EMM = +1.6 pps; $P < 0.001$, $d = 0.70$; $\sim$11.3% increase relative to baseline). In contrast, no significant changes were observed in the CON group. At recruitment, discharge rate slightly decreased from 7.90 [6.85, 8.96] to 7.80 pps [6.74, 8.87] ($\Delta$EMM = $-0.10$ pps; $P = 1.000$, $d = -0.06$), and during the plateau phase, from 15.2 [13.8, 16.6] to 15.1 pps [13.7, 16.4] ($\Delta$EMM = $-0.10$ pps; $P = 1.000$, $d = -0.04$). At derecruitment, *post hoc* comparisons revealed no significant within-group changes (all $P \geq 0.107$), and discharge rate remained unchanged between time points (INT: 5.83 [5.30, 6.36] *vs.* 6.17 pps [5.64, 6.71], $d = 0.35$; and CON: 6.04 [5.42, 6.66] *vs.* 5.86 pps [5.23, 6.48], $d = -0.18$).

As illustrated in Fig. 2*F*, increases in motor unit discharge were observed across participants in the INT

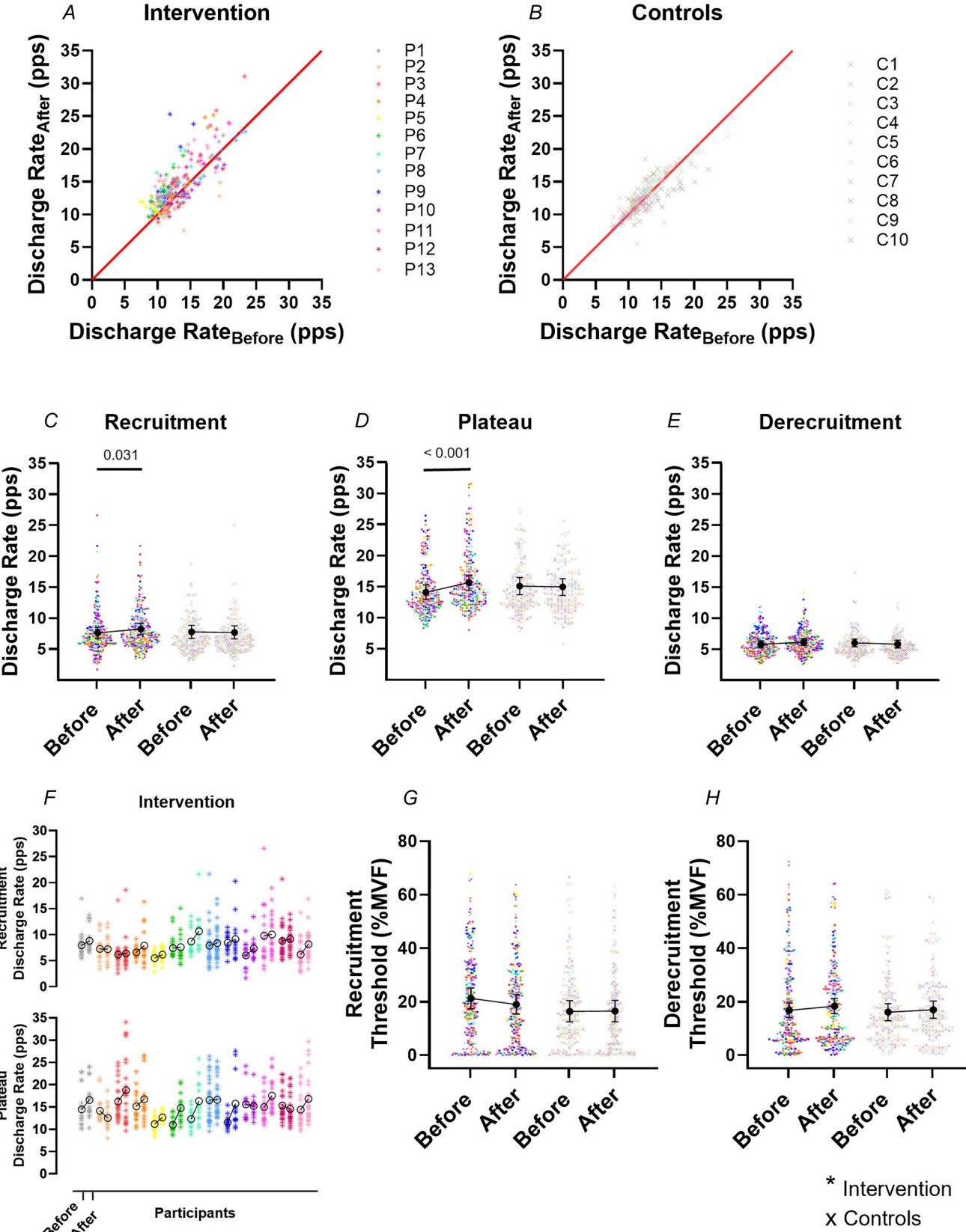

**Figure 2. Changes in motor unit properties following the strength training intervention**
*A* and *B*, scatter plots show discharge rates of the same motor units tracked from before to after the 4 week intervention for each participant in the intervention group (*A*, asterisks; *n* = 242) and control group (*B*, crosses;

*n* = 161). Each participant is represented by a unique colour across all panels. Average discharge rate values were computed over the entire trapezoidal contraction, and include all three contraction intensities [35%, 50% and 70% of maximal voluntary force (MVF)]. These values are shown before (*x*-axis) and after (*y*-axis) the intervention or control conditions. The red line indicates the line of identity. *C–E*, swarm plots show average motor unit discharge rates at recruitment (*C*), during the plateau (*D*) and at derecruitment (*E*) of the trapezoidal contractions for both groups before and after the intervention. Individual motor units are represented by asterisks (intervention) or crosses (controls) and clustered by participants. Data are collapsed across all contraction levels. In the intervention group, average motor unit discharge rate increased significantly both at recruitment (*P* = 0.031) and during the plateau (*P* < 0.001). *F* shows participant-specific changes (before *vs.* after) in average discharge rate at recruitment and during the plateau (*n* = 13). *G* and *H* show normalized (% MVF) recruitment and derecruitment thresholds for both groups before and after the intervention. Summary data in *C–E*, *G* and *H* are presented as estimated marginal means ± 95% confidence intervals; connection lines indicate the direction of change.

group following training, with 12 of 13 individuals showing an increase at recruitment and 10 of 13 during the plateau phase. Each asterisk represents a motor unit, colour-coded by participant, and black circles indicate the participant-level mean values. This individual-level pattern confirms the upward shift observed at the group level. To visualize whether these adaptations were consistent across the motor unit pool, changes in discharge rate were plotted against recruitment thresholds at 35%, 50% and 70% MVF (Fig. 3*A–C*). These plots, presented for illustrative purposes, show that increases in discharge rate occurred across the full range of recruitment thresholds, indicating that the training effect was not confined to early- or late-recruited motor units.

In contrast to discharge rate parameters, normalized motor unit recruitment [$\chi^2(1) = 0.87$, $P = 0.351$, $\eta^2_p = 0.04$] and derecruitment [$\chi^2(1) = 0.13$, $P = 0.721$, $\eta^2_p = 0.006$] thresholds were not significantly affected by the strength training intervention (Fig. 2*G* and *H*). Both recruitment (INT: 21.2% MVF [17.3, 25.1] *vs.* 18.9% MVF [15.3, 22.6], $d = -0.32$; and CON: 16.3% MVF [12.4, 20.3] *vs.* 16.4% MVF [12.4, 20.4], $d = 0.02$) and derecruitment thresholds (INT: 16.7% MVF [14.0, 19.4] *vs.* 18.3% MVF [15.5, 21.2], $d = 0.32$; and CON: 16.0% MVF [12.8, 19.2]

*vs.* 16.9% MVF [13.7, 20.2], $d = 0.17$) remained stable over time, indicating that a similar number of motor units was recruited to produce equivalent (normalized) submaximal forces before and after training.

To investigate whether the training-related increases in motor unit discharge rate contributed to strength gains, repeated-measures correlations were conducted between changes in ΔMVF and changes in discharge rate at both the recruitment and plateau phases (Fig. 4*A* and *B*). Results revealed large and statistically significant associations, indicating a positive within-participant relationship between changes in discharge rate and changes in muscle strength. Specifically, increases in discharge rate at recruitment were positively associated with ΔMVF ($r_{rm} = 0.57$, 95% CI [0.06, 0.85], $P = 0.033$; Fig. 4*A*), as were changes in discharge rate during the plateau phase ($r_{rm} = 0.54$, 95% CI [0.01, 0.83], $P < 0.05$; Fig. 4*B*).

### Estimate of persistent inward current amplitude

Motor unit pairs that met the inclusion criteria for Δ*F* analysis were identified successfully in all 10 CON participants and 13 INT participants during the 35% MVF

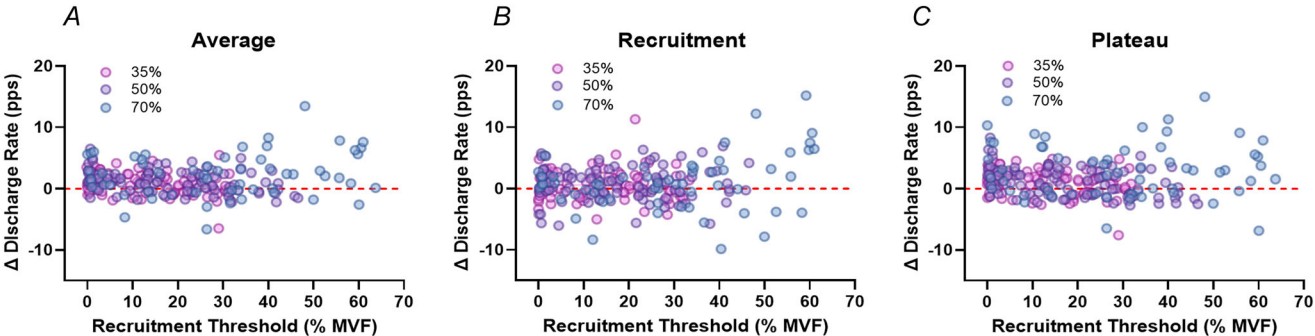

**Figure 3. Changes in motor unit discharge rate as a function of recruitment threshold**
Differences (Δ) in discharge rate of the tracked motor units before and after the training intervention are shown for each phase of the trapezoidal contractions, plotted relative to the normalized recruitment threshold after training [% maximal voluntary force (MVF)]. *A* shows the change in average discharge rate over the entire contraction; *B* shows the change at recruitment; and *C* shows the change during the plateau phase. Each data point represents a single motor unit identified at the corresponding force level (pink, 35% MVF; purple, 50% MVF; blue, 70% MVF). Data are presented for visualization purposes only, to illustrate the distribution of training-induced changes across the motor unit pool.

contractions, resulting in 99 unit-wise comparisons in the CON group and 117 in the INT group. At baseline, mean $\Delta F$ values were $4.75 \pm 1.57$ Hz in the CON group and $3.50 \pm 2.00$ Hz in the INT group. Following the training intervention, $\Delta F$ values increased in 10 of 13 participants in the INT group, whereas in the CON group, $\Delta F$ increased in half of the participants and decreased in the other half.

This pattern was reflected by a significant group × time interaction ($F_{1,129.87} = 11.49$, $P < 0.001$). *Post hoc* analysis confirmed a significant increase in $\Delta F$ in the INT group ($P < 0.001$, $d = 0.43$, 95% CI [0.17, 0.74], with an EMM difference of $+1.00$ Hz, 95% CI [0.46, 1.53]; Fig. 5A). No significant change was observed in the CON group ($P = 0.234$, $d = -0.18$, 95% CI [$-0.48$, 0.10]).

Additionally, repeated-measures correlation revealed that training-related changes in $\Delta F$ were positively associated with changes in motor unit discharge rate during the plateau phase of the contractions ($r_{rm} = 0.61$, $P = 0.021$; Fig. 5C). In contrast, no significant relationship was observed for the recruitment phase ($r_{rm} = 0.41$, $P = 0.148$; Fig. 5B).

### Comparative analysis of motor unit adaptability to strength training between young and older adults

To assess age-related differences in motor neuron adaptations, we compared the training-induced changes in motor unit properties observed in the present cohort of older adults with those previously reported in young adults who underwent the same 4 week strength training protocol (Del Vecchio et al., 2019).

The comparative analysis included 13 older adults and 12 young adults from the previous study, all belonging to the training groups, with 242 and 253 longitudinally tracked motor units, respectively, using identical decomposition and tracking procedures.

To account for interindividual variability in baseline performance (Wunderle et al., 2024), we verified whether baseline MVF differed between young and older adults; no significant association with age group was observed [$\chi^2(1) = 2.25$, $P = 0.133$; Fisher's exact test, $P = 0.187$], suggesting that baseline muscle strength did not clearly differentiate participants by chronological age.

Recruitment threshold (% MVF) decreased significantly following training [time effect: $\chi^2(1) = 10.29$, $P < 0.001$, $\eta^2_p = 0.24$], with no significant age × time interaction [$\chi^2(1) = 1.57$, $P = 0.211$, $\eta^2_p = 0.04$], indicating a similar pattern of reduction across age groups (Fig. 6A). Although the decrease appeared larger in young adults ($\sim$21.9%) than in older adults ($\sim$12.7%), the difference between groups was not statistically significant (Fig. 6A).

Discharge rate at recruitment increased following training [time effect: $\chi^2(1) = 10.29$, $P = 0.001$, $\eta^2_p = 0.17$] and was lower overall in older adults [age effect: $\chi^2(1) = 23.48$, $P < 0.001$, $\eta^2_p = 0.32$]. The absence of a significant age × time interaction [$\chi^2(1) = 0.48$, $P = 0.488$, $\eta^2_p = 0.01$] indicates a similar pattern of relative increase between age groups (Fig. 6B). In absolute terms, discharge rate at recruitment increased by $+0.68$ pps ($\sim$6.1%) in young adults and by $+0.57$ pps ($\sim$7.1%) in older adults.

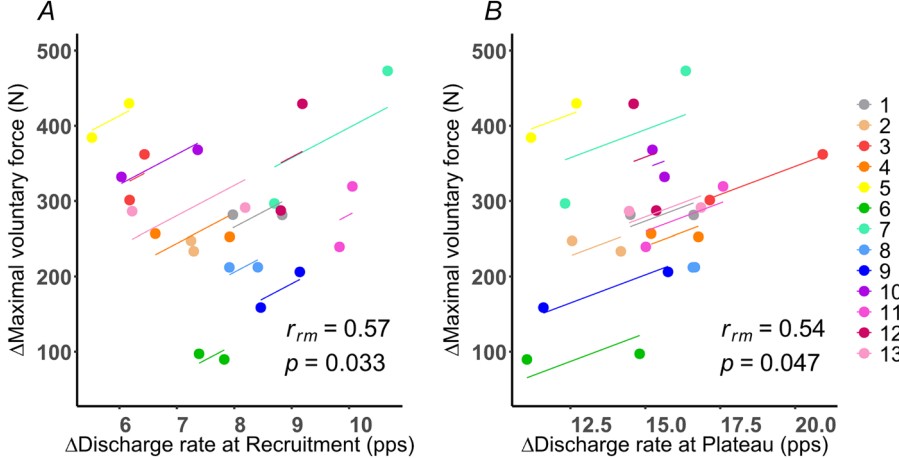

**Figure 4. Associations between changes in motor unit discharge rate and maximal voluntary force**
Repeated-measures correlation plots showing within-participant associations between changes ($\Delta$) in motor unit discharge rate at recruitment (*A*) and during the plateau phase (*B*) and changes in maximal voluntary force (MVF). Each coloured line represents the fitted regression line for an individual participant across the two time points (before and after training), with paired dots connected accordingly (lower dot = before training, upper dot = after training). Data are shown for the intervention group only ($n = 13$). The repeated-measures correlation coefficient ($r_{rm}$) and corresponding *P*-value are reported in the lower right corner of each panel.

In contrast, discharge rate during the plateau phase showed a significant age × time interaction [$\chi^2(1) = 10.8$, $P = 0.001$, $\eta^2_p = 0.13$], along with significant main effects of age [$\chi^2(1) = 43.7$, $P = 0.001$, $\eta^2_p = 0.38$] and time [$\chi^2(1) = 80.6$, $P = 0.001$, $\eta^2_p = 0.45$]. These results indicate that the training-induced increase in discharge rate differed between age groups, with greater adaptation in young adults (Fig. 6*C*). *Post hoc* analyses revealed significant increases in both groups; however, the increase was more pronounced in young adults (from 19.6 [18.3, 21.0] to 23.0 pps [21.6, 24.4]; $\Delta$EMM = +3.4 pps, +17.3%; $P < 0.001$, $d = 1.00$) compared with older adults (from 14.2 [12.8, 15.6] to 15.7 pps [14.3, 17.2]; $\Delta$EMM = +1.5 pps, +10.6%; $P < 0.001$, $d = 0.45$).

To provide a complementary, participant-level comparison across all subgroups, including the control groups from both studies, mean percentage changes ($\Delta$%) from pre- to post-training were calculated for each participant (Fig. 6*D*–*F*). Across recruitment threshold, discharge rate at recruitment and discharge rate during the plateau phase, two-way ANOVA revealed significant main effects of training status (no training *vs.* training, all $P < 0.05$), but no significant age × training interactions (all $P \geq 0.10$). These findings indicate that, when expressed as mean percentage change, both young and older adults exhibited similar training-related adaptations, whereas no systematic changes were observed in the untrained control groups.

Overall, these results demonstrate that the ageing neuromuscular system retains a substantial capacity for adaptation following short-term strength training, showing qualitatively similar changes in motor unit

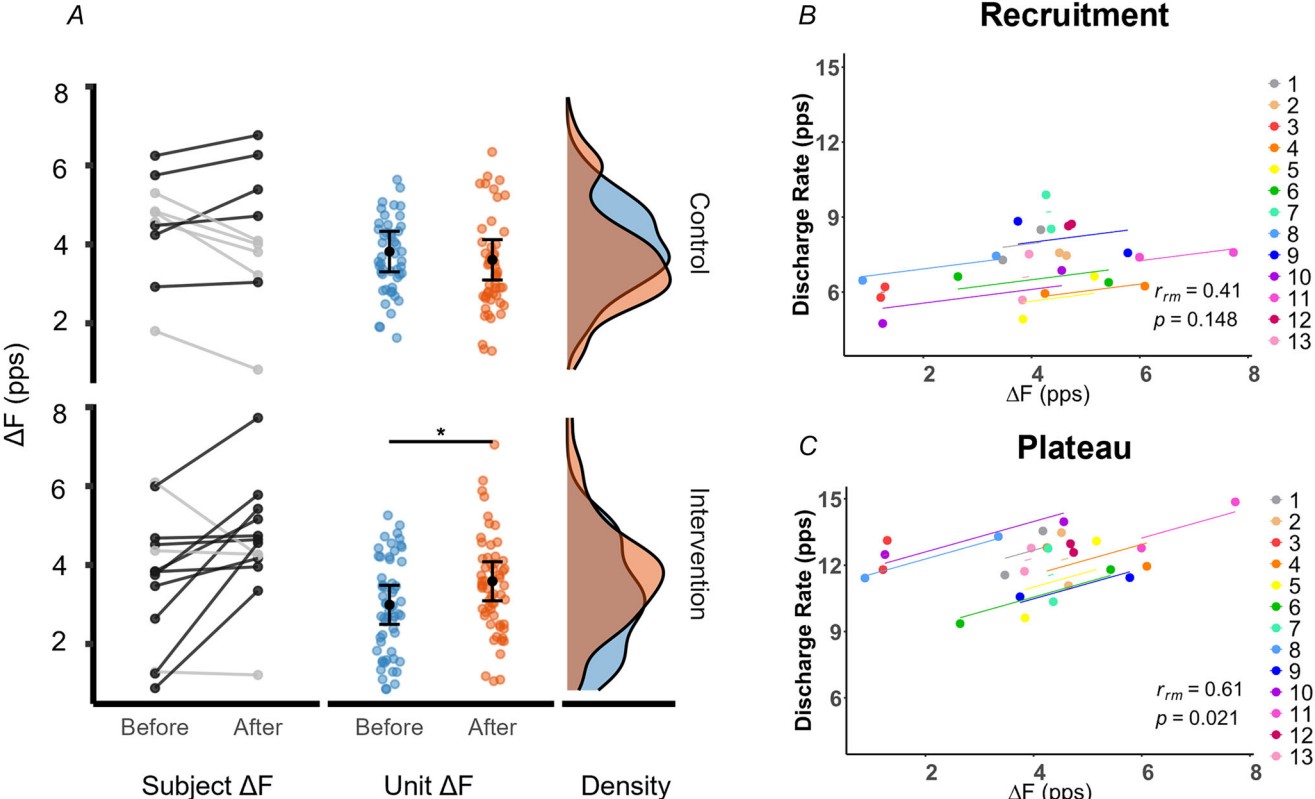

**Figure 5. Delta F during trapezoidal contractions at 35% of maximal voluntary force**
*A*, for each group, the left panel shows the average delta frequency ($\Delta F$) per participant before and after the 4 week strength training intervention, espressed in pulses per second (pps). Participants whose average $\Delta F$ increased are shown in black, whereas those whose average $\Delta F$ decreased are shown in grey. The middle panel displays unit-wise $\Delta F$ values per test unit. Estimated marginal means are indicated by black dots, with 95% confidence interval. Kernel density estimations are shown in the right panels (blue, before; orange, after). *B* and *C*, repeated-measures correlation plots illustrating the within-participant association between changes in $\Delta F$ and motor unit discharge rate at recruitment (*B*) and during the plateau phase (*C*) of contractions at 35% maximal voluntary force. Each coloured line represents the fitted regression for an individual participant across the two measurement sessions (before and after training), with paired dots connected accordingly (lower dot = before training, upper dot = after training). Data are shown for the intervention group only (*n* = 13). The repeated-measures correlation coefficient ($r_{rm}$) and corresponding *P*-values are reported in the lower right corner of each panel.

discharge behaviour and recruitment strategies to those observed in young adults exposed to the same training protocol.

## Discussion

In this study, we examined motor unit adaptations underlying the gains in muscle strength in older adults following a 4 week isometric strength training programme. A secondary aim was to compare these adaptations between young and older adults exposed to the same training programme. Using HDsEMG and longitudinal motor unit tracking, we assessed changes in discharge rate, recruitment threshold and estimates of PICs ($\Delta F$). Training induced a 17.6% increase in MVF, paralleled by a significant 11.3% increase in motor unit discharge rate during submaximal ramp contractions. In contrast, recruitment and derecruitment thresholds remained unchanged. Estimates of PICs increased after the training programme and were positively associated with changes in discharge rate, which, in turn, were correlated with improvements in maximal strength. Together, these findings indicate that aged motor neurons preserve the capacity for meaningful functional adaptation,

primarily through increased discharge rates during submaximal contractions and increased intrinsic excitability, adaptations that are qualitatively similar to those previously described in young adults exposed to the same training protocol (Del Vecchio et al., 2019).

### Motor unit adaptations to strength training in older adults

In older adults, motor unit activation during voluntary contraction is characterized by reduced discharge rates, reflecting age-related alterations in excitatory synaptic input, intrinsic motor neuron excitability and motor unit remodelling, which collectively impair volitional force production (Orssatto et al., 2021a, 2022; Piasecki et al., 2016; Watanabe et al., 2016). Despite these age-related alterations, accumulating evidence indicates that aged motor neurons preserve a capacity for adaptation in response to strength training. Within this physiological context, our results suggest that a short-term isometric strength training intervention can enhance motor neuron output in older adults, as reflected by increased discharge rates, both at recruitment (+8.2%) and during the plateau phase (+11.3%) of submaximal contractions.

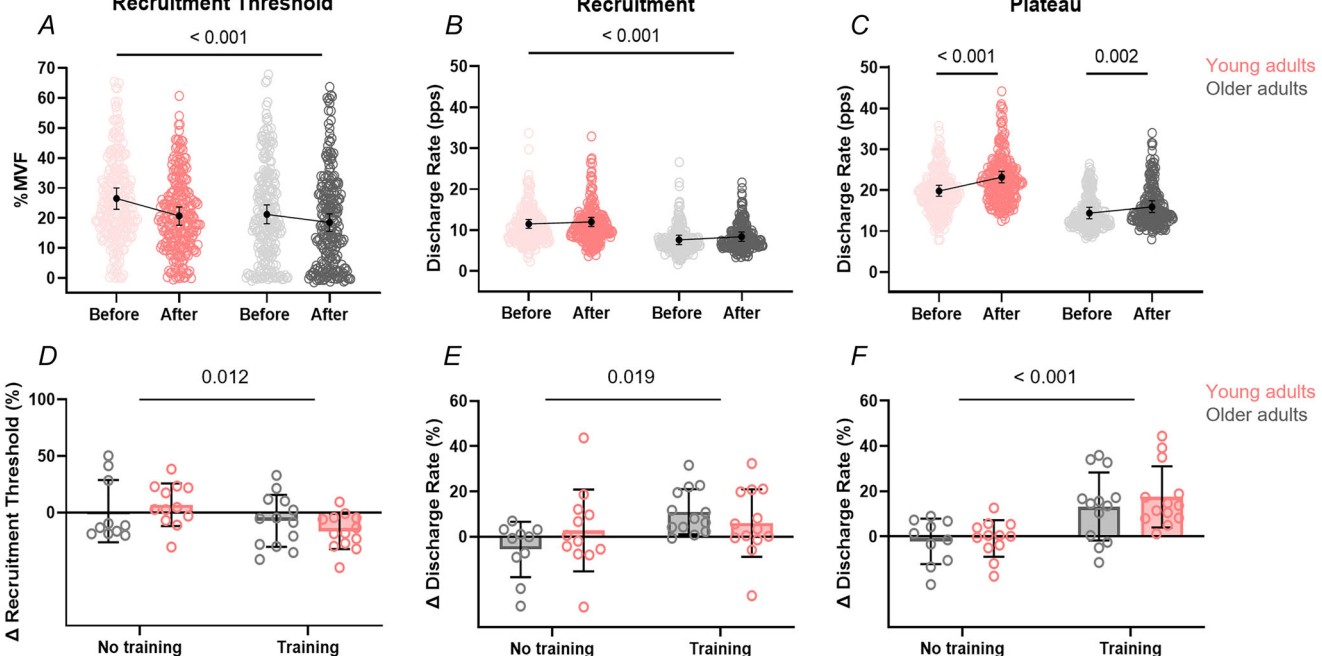

**Figure 6. Training-induced changes in motor unit properties in young and older adults**
*A–C*, swarm plots show motor unit recruitment threshold (*A*), discharge rate at recruitment (*B*) and discharge rate during the plateau phase (*C*) for young (pink) and older (grey) adults before and after the 4 week strength training programme. Each dot represents an individual motor unit. Black symbols and error bars indicate group estimated marginal means with 95% confidence intervals. Data are collapsed across all contraction levels. Reported *P*–values correspond to the results of the generalized linear mixed models (GLMM): significant main effect of time in *A* and *B*, and a significant group × time interaction in *C*. *D–F* show mean percentage changes (Δ%) in the same variables, calculated for each participant, including control groups from both studies (young and older adults). Bars represent group means ± SD. Statistical annotations refer to the main effect of training status from two-way ANOVAs.

Previous studies in older adults have primarily reported training-induced increases in motor unit discharge rate during maximal contractions. For instance, Kamen and Knight (2004) reported a 49% increase in maximal discharge rate from the first baseline to the post-training test, which was conducted after 6 weeks of dynamic training. However, no significant change was observed between the second baseline and post-test, suggesting that the increase occurred largely between the two baseline assessments (i.e. 1 week apart). This adaptation was confined to maximal contractions, with no changes at submaximal intensities. Likewise, Christie and Kamen (2010) observed a 24.3% increase in discharge rate during maximal efforts and a shortened after-hyperpolarization duration after 2 weeks of training. Together, these findings suggest that aged motor neurons remain responsive to training, particularly in conditions of high voluntary drive. Extending previous work, the present study demonstrates that motor neuron output in older adults can also be modulated at submaximal levels of activation. The absence of such adaptations in previous reports (Christie & Kamen, 2010; Kamen & Knight, 2004) is likely to reflect differences in training modality (dynamic *vs.* isometric) and task and training specificity (Giboin et al., 2018) or EMG methodology (intramuscular *vs.* HDsEMG). In this regard, the use of HDsEMG combined with longitudinal motor unit tracking enabled a more sensitive and consistent characterization of discharge rate adaptations in the same motor units over time, providing a more comprehensive estimate of the neural drive to muscle than intramuscular EMG (Farina et al., 2016; Lecce et al., 2026).

Consistent with recent evidence provided by Orssatto et al. (2023), we observed a small to moderate increase in PIC estimates ($\Delta F$, $d = 0.43$) following training. PICs amplify synaptic inputs and sustain motor neuron firing (Heckman et al., 2005). In line with this role, training-induced increases in $\Delta F$ were positively correlated with changes in motor unit discharge rate during the plateau phase (Fig. 5*C*), which, in turn, were associated with gains in MVF (Fig. 4*A* and *B*). Repeated-measures correlation analyses confirmed significant within-participant associations between $\Delta F$ and $\Delta$DR (delta discharge rate; $r_{rm} = 0.61$, $P = 0.021$) and between $\Delta$DR and $\Delta$MVF ($r_{rm} = 0.54$, $P = 0.047$), supporting the view that enhanced intrinsic motor neuron excitability facilitates a more effective translation of descending drive into motor neuron output in older adults (Lecce et al., 2025). Notably, no association was found between $\Delta F$ and discharge rate at recruitment, consistent with the notion that suprathreshold PICs, as estimated by the $\Delta F$ method, contribute minimally to the initial activation of motor neurons and are engaged primarily in sustaining firing during steady force production (Bennett et al., 1998; Heckman et al.,

2005; Mesquita et al., 2024). This pattern indicates that the largest training-induced adaptations in PICs are likely to occur at the suprathreshold level, enhancing self-sustained firing rather than influencing the initial discharge activity. Accordingly, the absence of a direct association between $\Delta F$ and MVF further supports the interpretation of $\Delta F$ as an index of self-sustained firing capacity rather than a direct measure of motor neuron gain or force output *per se*. In line with this interpretation, recruitment thresholds remained comparable before and after training, suggesting that common synaptic input and subthreshold PICs were largely unaffected (Lecce et al., 2026). Given the well-documented reduction in voluntary activation in older adults (Rozand et al., 2020), these suprathreshold adaptations might represent a key neural mechanism supporting initial strength gains in this population. Although the precise mechanisms underlying increased PICs strength in humans remain unclear, animal studies suggest that adaptations in monoaminergic input, particularly serotonergic and noradrenergic pathways, might enhance amplification of PICs by increasing motor neuron responsiveness (Behan et al., 2012; Orssatto et al., 2023). Strength training might also modify motor neuron biophysical properties, such as somato-dendritic excitability, as observed in rodent models (Krutki et al., 2017). Although speculative, these mechanisms provide a plausible physiological substrate for the observed increase in intrinsic excitability following training.

In contrast to discharge behaviour, motor unit recruitment or derecruitment thresholds did not change following training. In older adults, evidence regarding training-induced adaptations in recruitment strategies remains inconsistent. Orssatto et al. (2023) reported increased recruitment thresholds after 6 weeks of training, suggesting that resistance training might help to restore the ability to rely more on discharge rate modulation rather than early recruitment. This interpretation aligns with previous observations of lower recruitment thresholds and slower discharge rates in older adults, which might reflect a compensatory response to reduced motor neuron excitability (Orssatto et al., 2021a, 2022). Within this context, the enhanced discharge rate observed in the present study, particularly during the recruitment phase, might indicate a shift towards greater rate modulation at the onset of activation. However, recruitment thresholds represent a complex and indirect measure influenced by multiple factors beyond motor neuron properties, including antagonist co-activation (De Luca & Mambrito, 1987) and the cumulative force generated by previously recruited units, and should therefore be interpreted with caution. The absence of recruitment threshold adaptations in our study might also reflect the relatively short duration of the intervention, the task specificity of the isometric protocol, or might indicate that changes in discharge behaviour and intrinsic

excitability represent the primary neural mechanisms supporting initial strength gains in older adults.

The strength gains observed in the present study (+17.6%) align with previous reports of short-term training adaptations in older adults. For instance, Kamen and Knight (2004) and Orssatto et al. (2023) reported increases of +36% and +25%, respectively, after 6 weeks of dynamic or power-oriented training. The slightly smaller gain in our study might reflect the shorter duration (4 *vs*. 6 weeks) and use of an isometric protocol. In contrast, Christie and Kamen (2010) reported a comparable improvement (+13%) following 2 weeks of isometric strength training. Collectively, these findings indicate that short-term, task-specific strength training can elicit meaningful improvements in MVF in older adults, mediated, at least in part, by early motor neuron-level adaptations. Although antagonist muscle co-activation was not assessed directly in the present study, its potential contribution to net force output cannot be excluded entirely. However, available evidence suggests that changes in co-activation contribute minimally to strength gains in older adults, particularly in short-term interventions (Walker, 2021). Reductions in antagonist activity have been observed inconsistently and primarily in individuals with unusually high baseline co-activation levels (Häkkinen et al., 1998, 2000). Moreover, antagonist activation might serve a stabilizing role at the knee joint level (Baratta et al., 1988), especially in older individuals, which might limit its adaptability in response to training.

Although muscle hypertrophy was not assessed directly in the present study, its occurrence appears unlikely given the short duration of the intervention and the use of low-volume isometric training. Supporting this assumption, Orssatto et al. (2023) reported no detectable hypertrophy, measured as muscles thickness and cross-sectional area, of the lower-limb musculature after a 6 week power-oriented resistance training protocol. If structural changes were absent in those conditions, they are arguably even less likely following a shorter, lower-volume, isometric training intervention. Accordingly, the strength gains observed in our study can be attributed primarily to neural adaptations, which typically dominate the early phase of strength training (Pearcey et al., 2021; Škarabot et al., 2021) and offer a valuable window into the neural mechanisms supporting force gains in older adults.

## Comparison of motor unit adaptations between young and older adults

This study provides the first direct comparison of motor unit adaptations to an identical short-term strength training protocol in young and older adults using HDsEMG and longitudinal motor unit tracking. This matched approach enabled the assessment of age-related differences in motor neuron adaptations in controlled experimental conditions, allowing for a mechanistic comparison of training-induced motor neuron changes across the adult lifespan.

Overall, both age groups exhibited clear strength training-induced adaptations. However, the expression of these adaptations was not identical. The young adults exhibited a reduction in recruitment threshold without a concomitant increase in discharge rate at recruitment, whereas older adults displayed a distinct pattern characterized by a significant increase in discharge rate at motor unit onset. Although recruitment threshold did not change significantly in older adults compared with age-matched control subjects (Fig. 2*G*), the direction of change was similar to that observed in young adults, albeit with a smaller magnitude (−12.7% *vs*. −21.9%), suggesting a partial but attenuated modulation of recruitment strategy (Fig. 6*A* and *B*). Notably, older adults exhibited a significant increase in discharge rate at recruitment, whereas no such change was observed in young adults after the same training protocol (Del Vecchio et al., 2019). This finding suggests an age-specific pattern of neural adaptation, whereby older adults might increase discharge rate at motor unit recruitment when modulation of recruitment threshold is limited. Such a response might represent a compensatory strategy to enhance force contribution at motor unit onset, reflecting an alternative means of increasing motor neuron output when recruitment modulation is constrained by ageing-related reductions in excitability or altered synaptic integration (Barry et al., 2007), although the precise mechanisms underlying this adaptation remain to be determined.

The only parameter showing a significant group × time interaction was motor unit discharge rate during the plateau phase of submaximal contractions, which increased in both groups (Fig. 6 *C*), but to a greater extent in young adults (+16.9% *vs*. +11.3%). This phase reflects the capacity to increase and sustain motor neuron firing during constant-force output, which relies on both intrinsic motor neuron excitability and effective central drive. Although the ageing motor neuron retains a robust capacity for adaptation, the smaller increase observed in older adults suggests that rate coding during sustained force production might be less responsive to training with advancing age. This attenuated adaptation is consistent with evidence from acute studies showing age-related differences in neuromuscular responses following resistance training. For instance, Nishikawa et al. (2024) reported a reduced ability of older adults to modulate motor unit discharge rate after an acute bout of resistance exercise, particularly in relationship to recruitment threshold. Together with previous observations of reduced motor neuron excitability and impaired central drive in

older individuals (Klass et al., 2008; Orssatto et al., 2022), these findings suggest that ageing-related constraints in sustaining high-frequency discharges might limit the magnitude of training-induced adaptations during the plateau phase. Nevertheless, the consistent direction of change across age groups indicates that older adults can still engage similar neural strategies to enhance force production in response to strength training, albeit to a lesser extent than young adults.

A clustering analysis based on baseline muscle strength (i.e. pretraining MVF) did not differentiate participants by age group, indicating that differences in initial strength capacity alone were not sufficient to account for the observed age-related differences in motor unit adaptations.

## Conclusion

In this study, we examined whether the motor unit adaptations contributing to initial strength gains are preserved in older adults undergoing a short-term iso-metric strength training intervention. Our findings demonstrate that, despite age-related reductions in base-line motor neuron excitability and discharge rate, the ageing neuromuscular system retains a robust capacity for functional adaptation. Specifically, we observed training-induced increases in motor unit discharge rate and estimates of intrinsic excitability, which were associated with improvements in MVF. Although some aspects of discharge rate, particularly during the plateau phase of voluntary contractions, appear attenuated with age, the overall pattern of adaptation is qualitatively similar to that observed in young adults exposed to the same training protocol. These results underscore the relevance and efficacy of strength training in promoting neural adaptations in older individuals and highlight the preserved potential of aged motor neurons to support gains in voluntary force.

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

## Additional information

### Data availability statement

All data supporting the findings of this study are available from the corresponding author upon reasonable request.

### Competing interests

The authors declare that they have no competing interests.

### Author contributions

A.C., S.D.V., J.S., D.F. and A.D.V. conceived and designed research. A.C., S.D.V., B.S. and A.D.V. performed experiments. A.C., S.D.V., B.G., S.N., E.L., L.A. and A.D.V. analysed data. A.C., S.D.V., B.G., S.N., E.L., L.A., I.B., F.F., J.S., D.F. and A.D.V. interpreted results of experiments. A.C. prepared the figures and drafted the manuscript. All authors revised the manuscript critically for important intellectual content. All authors have read and approved the final version of manuscript and agree to be accountable for all aspects of the work in ensuring that questions related to the accuracy or integrity of any part of the work are appropriately investigated and resolved. All persons designated as authors qualify for authorship, and all those who qualify for authorship are listed.

### Funding

This work was supported by European Research Council (ERC) grant 101118089 (A.D.V.), German Ministry for Education and Research (BMBF) grants 01DN2300 and 16SV9246 (A.D.V.), German Research Foundation (DFG) grant 523352235 (A.D.V.), Bavarian Ministry of Economic Affairs, Regional Development and Energy (StMWi) grant MV-2303-0006 (A.D.V.), and Bavarian Ministry of Economic Affairs, Regional Development and Energy (StMWi) grant LSM-2303-0003 (A.D.V.).

### Acknowledgements

We would like to thank all the Volunteers for their time and efforts in completing the study.

### Keywords

ageing, high-density surface EMG, motor neuron, motor unit, strength training

## Supporting information

Additional supporting information can be found online in the Supporting Information section at the end of the HTML view of the article. Supporting information files available:

**Peer Review History**

