## [Peer Review History · The Journal of Physiology]

Ageing Does Not Impair Motor Neuron Adaptations: Comparable Motor Unit Responses to Strength Training in Young and Older Adults

Andrea Casolo, Stefanie Del Vecchio, Benjamin Goodlich, Bastian Schrader, Stefano Nuccio, Edoardo Lecce, Ilenia Bazzucchi, Luca Angius, Francesco Felici, Joachim Schrader, Dario Farina, and Alessandro Del Vecchio
DOI: 10.1113/JP290541

Corresponding author(s): Alessandro Del Vecchio (alessandro.del.vecchio@fau.de)

The following individual(s) involved in review of this submission have agreed to reveal their identity: Kohei Watanabe (Referee #2)

Review Timeline:	Submission Date:	13-Nov-2025
	Editorial Decision:	03-Dec-2025
	Revision Received:	04-Feb-2026
	Accepted:	10-Feb-2026

Senior Editor: Karyn Hamilton

Reviewing Editor: Kevin Murach

Transaction Report:

Re: JP-RP-2025-290541 "**Ageing Does Not Impair Motor Neuron Adaptations: Comparable Motor Unit Responses to Strength Training in Young and Older Adults**" by Andrea Casolo, Stefanie Del Vecchio, Benjamin Goodlich, Bastian Schrader, Stefano Nuccio, Edoardo Lecce, Ilenia Bazzucchi, Luca Angius, Francesco Felici, Joachim Schrader, Dario Farina, and Alessandro Del Vecchio

Dear Dr Del Vecchio,

Thank you for submitting your manuscript to The Journal of Physiology. It has been assessed by a Reviewing Editor and by 2 expert referees and we are pleased to tell you that it is potentially acceptable for publication following satisfactory major revision.

Please address all the points raised and incorporate all requested revisions or explain in your Response to Referees why a change has not been made. We hope you will find the comments helpful and that you will be able to return your revised manuscript within 2 months. If your article is NOT for a Special Issue, you may have 9 months to revise. If you require an extension, please contact journal staff: jp@physoc.org. Please note that this letter does not constitute a guarantee for acceptance of your revised manuscript.

REVISION CHECKLIST:

Upload a full Response to Referees file. To create your 'Response to Referees': copy all the reports, including any comments from the Senior and Reviewing Editors, into a Microsoft Word, or similar, file and respond to each point, using

font or background colour to distinguish comments and responses and upload as the required file type.

We look forward to receiving your revised submission.

Yours sincerely,

Karyn Hamilton
Senior Editor
The Journal of Physiology

REQUIRED ITEMS

1) - Author photo and profile. First or joint first authors are asked to provide a short biography (no more than 100 words for one author or 150 words in total for joint first authors) and a portrait photograph. These should be uploaded and clearly labelled together in a Word document with the revised version of the manuscript. See Information for Authors for further details.

2) - The contact information for the person responsible for 'Research Governance' at your institution needs to be provided. This includes their name and an institutional email address. Please ensure the contact is not an author on this paper and provide an alternate contact if necessary, or confirm in the submission form that the author whose email was provided has sole responsibility for research governance. This is the person who is responsible for regulations, principles and standards of good practice in research carried out at the institution, for instance the ethical treatment of animals, the keeping of proper experimental records or the reporting of results.

3) - You must start the Methods section with a paragraph headed Ethical Approval. If experiments were conducted on humans, confirmation that informed consent was obtained, preferably in writing, that the studies conformed to the standards set by the latest revision of the Declaration of Helsinki and that the procedures were approved by a properly constituted ethics committee, which should be named, must be included in the article file. If the research study was registered (clause 35 of the Declaration of Helsinki), the registration database should be indicated, otherwise the lack of registration should be noted as an exception (e.g. The study conformed to the standards set by the Declaration of Helsinki, except for registration in a database). For further information see: <https://physoc.onlinelibrary.wiley.com/hub/human-experiments>.

4) - Please upload separate high-quality figure files via the submission form.

5) - Papers must comply with the Statistics Policy: https://jp.msubmit.net/cgi-bin/main.plex?form_type=display_requirements#statistics.

In summary:

- If n {less than or equal to} 30, all data points must be plotted in the figure in a way that reveals their range and distribution. A bar graph with data points overlaid, a box and whisker plot or a violin plot (preferably with data points included) are acceptable formats.

- If $n > 30$, then the entire raw dataset must be made available either as supporting information, or hosted on a not-for-profit repository, e.g. FigShare, with access details provided in the manuscript.

- 'n' clearly defined (e.g. x cells from y slices in z animals) in the Methods. Authors should be mindful of pseudoreplication.
- All relevant 'n' values must be clearly stated in the main text, figures and tables.
- The most appropriate summary statistic (e.g. mean or median and standard deviation) must be used. Standard Error of the Mean (SEM) alone is not permitted.
- Exact p values must be stated. Authors must not use 'greater than' or 'less than'. Exact p values must be stated to three significant figures even when 'no statistical significance' is claimed.

6)- Please include an Abstract Figure file and an Abstract Figure legend. An appropriate figure legend, which should not exceed 150 words in length, should be included in the main manuscript file. The Abstract Figure is a piece of artwork designed to give readers an immediate understanding of the research and should summarise the main conclusions. If possible, the image should be easily 'readable' from left to right or top to bottom. It should show the physiological relevance of the manuscript so readers can assess the importance and content of its findings. Abstract Figures should not merely recapitulate other figures in the manuscript. Please try to keep the diagram as simple as possible and without superfluous information that may distract from the main conclusion(s). Abstract Figures must be provided by authors no later than the revised manuscript stage and should be uploaded as a separate file during online submission labelled as File Type 'Abstract Figure'. Please also ensure that you include the figure legend in the main article file. All Abstract Figures should be created using BioRender. Authors should use The Journal's premium BioRender account to export high-resolution images. Details on how to use and access the premium account are included as part of this email.

7)- Please include a full title page as part of your main article (Word) file, which should contain the following: title, authors, affiliations, corresponding author name and contact details, keywords, and running title.

EDITOR COMMENTS

Senior Editor:

Thank you for submitting your manuscript for consideration by The Journal of Physiology. As part of the peer review process, we recruited two Referees with expertise in this field of study. Each Referee provided detailed feedback including some major concerns that limit the impact of the manuscript in its current form. At this point, if you believe you can adequately address the critiques, we would like to invite you to respond point-by-point to each Referee comment, making corresponding revisions to your manuscript. We look forward to receiving your revised manuscript and thank you for your interest in The Journal of Physiology.

Reviewing Editor:

Your work has been evaluated by two experts in the field. Both agree that the work is interesting and the comments seem addressable. Reviewer 1 raised concerns about the heterogeneity that can occur with aging. It will be particularly important to address the following: "Does a cluster analysis of the combined MVF values for both cohorts, for example, yield two groups that differ only in chronological age? If not, caution is necessary when comparing the two groups." Reviewer #2 requested more Discussion to place this work in better context with the literature. Reviewer 2 also raises a valid point about the contribution of antagonistic muscles with aging. Please respond to the comments in full.

REFEREE COMMENTS

Referee #1:

The study examined the capacity of older adults to exhibit motor unit adaptations to short-term strength training previously shown to enhance neural drive to muscle in young adults. The study enrolled 23 older adults who were assigned to either an intervention group (71{plus minus} 4 yrs) or a control group (69{plus minus} 2 yrs). The training program (rapid and sustained isometric contractions) comprised three sessions per week for 4 weeks. Participants in the Control group were instructed to maintain habitual levels of physical activity but should have been given a non-strength training intervention. Motor unit activity in tibialis anterior was recorded with HDsEMG electrodes. The main findings of the study an increase in MVF that

was accompanied by an increase in motor unit discharge rate during the submaximal isometric contractions. The results do not indicate the relative contributions of the outcome variables to the increase in the strength of the dorsiflexor muscles.

Major Comments

1. Due to the existence of low and high performers across the lifespan (Wunderlee et al. *Front Aging Neurosci* 16 1368052, 2024), caution is necessary when comparing groups of young and older adults. The implicit assumption of such comparisons is that the only difference between groups is chronological age. Given that the reported data are compared with those obtained in previous study on young adults (line 574), it is necessary to demonstrate that the only difference between the two cohorts (young and old) is indeed chronological age. Does a cluster analysis of the combined MVF values for both cohorts, for example, yield two groups that differ only in chronological age? If not, caution is necessary when comparing the two groups.

2. Although the protocol included a single familiarization session, are you certain that this was sufficient for the participants to learn the task. For example, Kamen & Knight (*J Gerontol Med Sci* 59A; 1334, 2005) had participants perform two baseline tests one week apart before beginning a 6-week strength-training program with the knee extensors. MVC force increased for both young and older adults across the two baseline tests and the end of the 6-week intervention. In contrast, peak discharge rate of motor units in vastus lateralis during MVCs increased across the two baseline visits for both groups but there were no further increases after training. Moreover, the correlation between the increase in MVC force and change in discharge rate at the end of the 6-week intervention was 0.4 for the older adults. Was the current study able to distinguish between learning the MVC task and the adaptations evoked by strength training. For example, did you measure the outcome variables during visit 1 (familiarization) for comparison with those after the habituation activities (visit 2)?

3. The average increase in the strength of the dorsiflexor muscles was 17.6% after the 4-week intervention, which is somewhat less than the 36% increase reported by Kamen & Knight (2004) for the knee extensors after a 6-week intervention. The current study reported an average increase in discharge rate of 11.3% during the plateau phase of the submaximal trapezoidal contractions, although the target force is not specified. In contrast, Kamen & Knight (2004) found no statistically significant change in discharge rate relative to the second baseline test during MVCs performed after the 6-week intervention. It is also puzzling that the current study reported a significant correlation ($r = 0.54$) between the changes in MVF and discharge rate during the submaximal trapezoidal contractions. The differences between these two studies needs to be reconciled.

4. Were the estimates of persistent inward currents correlated with the increase in MVF?

5. If it is possible, develop a multiple-regression model to identify the outcome variables that can explain the increase in MVF.

Minor Comments

70 State the test muscle.

199 Define instantaneous.

470 All three target forces?

574 This comparison requires evidence that the only difference between the two groups was chronological age, as implied.

631 Define "enhanced discharge behaviour".

637 This sentence is misleading; report the changes after the two baseline sessions.

Referee #2:

The present study investigated motor unit adaptations underlying strength gains following a four-week isometric strength training program in older adults. The authors completed excellent works, and this knowledge would contribute to various research area and exercise or rehabilitation-associated fields. However, some part needs more detailed physiological considerations. This version just reported the results of older adults and comparison with the results of young adults. More discussion would help to make this article higher scientific value.

Abstract

L78-80

This part would induce misunderstanding that the present study used both young and older groups. The authors must not describe the results of young adults in the abstract.

L108-115

Some previous studies would reported characteristic motor unit activation pattern during acute motor tasks (ex. Nishikawa et al. *Exp Gerontol* 181: 112283, 2023). These findings help the authors to assume or hypothesize the characteristic age-related neural adaptation following chronic exercise intervention.

L123

If the authors include the intervention studies with nutritional supplementations, there are various reports that the motor unit adaptations in older adults following resistance training (Nishikawa et al. *Eur J Nutr* 64: 117, 2025, Watanabe et al. *Journal of Gerontology Series A: Biological Sciences* 75(5): 867-874, 2020). "Scarce" would be insufficient to review the related research.

L145

The descriptions of characteristic neural excitability in older adults were not fully reviewed in the Introduction. Also, the reason that the authors hypothesize similar pattern with young adults is unclear. Please describe physiological explanation for this hypothesis.

L334

How did the authors control the effects of antagonistic muscles? The older adults generally show greater antagonistic muscle activities.

L402-412

Figure 1 is repeatedly used in the Methods and Results. It would be better to use once in Methods or Results.

Figure 2 A and B

Individual plots are difficult to identify. How many plots did include in this figure? Smaller plot markers might help to show all plots.

Figures 4 and 5BC

Means of two plots within a participant were not described, i.e., Before and After. We can assume that left lower point would be Before and another could be After. However, it would be better to show it.

L574

Brief descriptions of detailed data set should be stated in this part, i.e., number of tracked MUs for young adults.

L635-716

As pointed in hypothesis, the authors didn't consider the age-related physiological responses or physiological conditions. This part of Discussion would be also able to be written from characteristics of older adults. Description that the results are similar with previous studies or the studies in young adults could be important, but comparisons with young adults are well documented in next session. Therefore, this part can be focused on the discussion for age-related physiological responses. Since baseline of motor unit activation properties should be affected in older adults, the authors must start from the summary of motor unit firing properties in older adults.

L724-735

Although similar responses were detected between young and older adults, detailed responses were not same. These differences can be detected by motor unit decomposition with HD-SEMG and strength point of this study. Thus, the authors must not conclude that older adults showed similar adaptations with young adults. Please add the discussion for the differences in recruitment threshold and/or a significant increase in discharge rate at recruitment.

L738-745

More physiological discussion should be needed for this part. Why were this difference observed in plateau phase? As pointed above, the authors can cite some research that showed difference in acute response of neuromuscular activation between young and older adults.

END OF COMMENTS

EDITOR COMMENTS

Senior Editor:

Thank you for submitting your manuscript for consideration by The Journal of Physiology. As part of the peer review process, we recruited two Referees with expertise in this field of study. Each Referee provided detailed feedback including some major concerns that limit the impact of the manuscript in its current form. At this point, if you believe you can adequately address the critiques, we would like to invite you to respond point-by-point to each Referee comment, making corresponding revisions to your manuscript. We look forward to receiving your revised manuscript and thank you for your interest in The Journal of Physiology.

REPLY: We thank the Senior Editor for the opportunity to revise our manuscript and for the constructive and detailed feedback provided by the two Referees. We have carefully considered all comments and have addressed each of the major concerns raised. In the revised manuscript, we have implemented substantial revisions to improve the physiological interpretation of the findings, clarify age-related comparisons, and strengthen the discussion of motor unit level mechanisms underlying observed strength gains. All changes have been made in direct response to the Referees' suggestions and are detailed in the point-by-point responses provided below. We believe that these revisions have significantly improved the clarity, rigor, and impact of the manuscript.

Reviewing Editor:

Your work has been evaluated by two experts in the field. Both agree that the work is interesting and the comments seem addressable. Reviewer 1 raised concerns about the heterogeneity that can occur with aging. It will be particularly important to address the following: "Does a cluster analysis of the combined MVF values for both cohorts, for example, yield two groups that differ only in chronological age? If not, caution is necessary when comparing the two groups." Reviewer #2 requested more Discussion to place this work in better context with the literature. Reviewer2 also raises a valid point about the contribution of antagonistic muscles with aging. Please respond to the comments in full.

REPLY: We thank the Reviewing Editor for the evaluation of our manuscript and for the constructive comments provided. We appreciate the positive assessment of our work and the opportunity to address the points raised by the Referees. In the revised manuscript, we have explicitly addressed the concern (Major comment - 1) raised by Reviewer 1 regarding heterogeneity across the lifespan by performing a clustering analysis of baseline maximal voluntary force values pooling young and older adults. This analysis demonstrated that baseline strength did not cluster participants by age group, and we have incorporated these results into both the Results and Discussion to appropriately contextualize age-related comparisons. In response to Reviewer 2, we have substantially expanded the Discussion to better integrate our findings within the existing literature on motor unit adaptations in older adults, including age-related physiological mechanisms and comparisons with previous acute and chronic studies. We have also explicitly addressed the potential contribution of antagonist muscle activity with ageing, discussing its likely influence based on available evidence. We believe that these revisions have strengthened the physiological interpretation and overall clarity of the manuscript. A detailed, point-by-point response to all comments is provided below.

REFEREE COMMENTS

Referee #1:

The study examined the capacity of older adults to exhibit motor unit adaptations to short-term strength training previously shown to enhance neural drive to muscle in young adults. The study enrolled 23 older adults who were assigned to either an intervention group (71±4 yrs) or a control group (69±2 yrs). The training program (rapid and sustained isometric contractions) comprised three sessions per week for 4 weeks. Participants in the Control group were instructed to maintain habitual levels of physical activity but should have been given a non-strength training intervention. Motor unit activity in tibialis anterior was recorded with HDsEMG electrodes. The main findings of the study an increase in MVF that was accompanied by an increase in motor unit discharge rate during the submaximal isometric contractions. The results do not indicate the relative contributions of the outcome variables to the increase in the strength of the dorsiflexor muscles.

REPLY: We thank the reviewer for the thorough and constructive evaluation of our manuscript. As suggested, we have performed additional analyses to explicitly address inter-individual variability in baseline performance (Major comment - 1), clarified the distinction between task familiarization and training-induced adaptations (Major comment - 2), and integrated the discussion to better contextualize the observed motor unit adaptations with respect to previous studies (major comment 3). We have also refined the interpretation of associations between neural variables and strength gains, explicitly acknowledging methodological and statistical constraints where needed (Major comment 4 and 5). We believe that these revisions have substantially improved the clarity and robustness of our manuscript.

Major Comments

1. Due to the existence of low and high performers across the lifespan (Wunderlee et al. Front Aging Neurosci 161368052, 2024), caution is necessary when comparing groups of young and older adults. The implicit assumption of such comparisons is that the only difference between groups is chronological age. Given that the reported data are compared with those obtained in previous study on young adults (line 574), it is necessary to demonstrate that the only difference between the two cohorts (young and old) is indeed chronological age. Does a cluster analysis of the combined MVF values for both cohorts, for example, yield two groups that differ only in chronological age? If not, caution is necessary when comparing the two groups.

REPLY: We thank the reviewer for this important observation and agree that inter-individual variability in performance may complicate age-based comparisons, as in our case (Young vs. older adults). To address this concern, as suggested, we performed a K-means clustering analysis on baseline MVF values pooling young and older adults. Thereafter, we tested whether the cluster solution aligned with age group identity (young vs. older adults) using both a chi-square test and Fisher's exact test. The results indicated no statistically significant association between the strength-based clusters and chronological age group ($\chi^2 = 2.25$, $p = 0.133$; Fisher's exact test, $p = 0.187$). Therefore, participants from both age groups were distributed across both strength-based clusters.

This cluster analysis suggests that baseline force level alone did not clearly separate older from younger participants (i.e. participants were not separated into low- and high-performers based on age), and that group differences observed in motor unit adaptations cannot be attributed solely to differences in initial strength. This analysis provides additional support for the interpretation of our group-based comparisons, while acknowledging that age-related adaptations may arise through multiple interacting factors beyond chronological age or baseline strength level alone.

Importantly, we wish to emphasize that the primary focus of this study is on motor unit adaptations rather than on maximal voluntary force per se. MVF served as a functional outcome to

contextualize the neural findings, but the core conclusions rest on training-induced changes in motor unit discharge rate and estimates of persistent inward currents (ΔF). These neural variables showed internally consistent and correlated adaptations within the older cohort i.e., discharge rate changes correlated with ΔF changes, which in turn correlated with MVF gains, independent of whether individual participants would be classified as high or low performers based on baseline force output. Therefore, even in the presence of inter-individual variability in initial strength, the interpretation of preserved neural adaptability in older adults remains well supported by the motor unit-level data

This analysis has now been integrated into the Results section (LL 599-602) and Discussion (LL 793-795) to appropriately contextualize age-related comparisons.

2. Although the protocol included a single familiarization session, are you certain that this was sufficient for the participants to learn the task. For example, Kamen & Knight (J Gerontol Med Sci 59A; 1334, 2005) had participants perform two baseline tests one week apart before beginning a 6-week strength-training program with the knee extensors. MVC force increased for both young and older adults across the two baseline tests and the end of the 6-week intervention. In contrast, peak discharge rate of motor units in vastus lateralis during MVCs increased across the two baseline visits for both groups but there were no further increases after training. Moreover, the correlation between the increase in MVC force and change in discharge rate at the end of the 6-week intervention was 0.4 for the older adults. Was the current study able to distinguish between learning the MVC task and the adaptations evoked by strength training. For example, did you measure the outcome variables during visit 1 (familiarization) for comparison with those after the habituation activities (visit 2)?

REPLY: We thank the reviewer for the comment. It is true that we did not record the main outcome variables (i.e. MVF and HDsEMG data) during the familiarization session (visit 1). However, all baseline measures used in our analysis were collected ~1 week later (visit 2), after participants had been habituated to the experimental procedures and had practiced both MVCs and submaximal trapezoidal contractions. This approach is consistent with previously adopted protocols (Del Vecchio et al., J Physiol, 2019; Casolo et al., J Electromyogr Kinesiol, 2020), ensuring that the data used for pre- vs. post- intervention comparisons were acquired once participants had become accustomed to the task.

Importantly, our study included a non-training control group (n = 10), which underwent the same pre- and post-testing procedures as the intervention group but did not perform the training. This group showed no changes in MVC or motor unit properties, indicating that learning effects or test-retest variability are unlikely to explain the adaptations observed in the intervention group.

It is also worth noting that the study by Kamen & Knight did not include a non-training control group for either young or older adults. As such, the authors likely implemented two baseline assessments to account for potential learning effects within each group. Furthermore, motor unit activity in the mentioned study was recorded using intramuscular EMG, which does not allow for longitudinal tracking of the same units, and is limited to the sampling of a small number of active motor units (i.e. high selectivity; Merletti et al., J Electromyogr Kinesiol, 2008). In contrast, the use of HDsEMG in the present study allowed the identification of a larger and more representative pool of motor units, which could be longitudinally tracked within each participant between testing sessions, thereby enhancing the resolution and sensitivity to detect changes in motor unit properties.

While we acknowledge that recording outcome variables during the familiarization visit would have provided an additional level of control, we consider that the combination of an in-depth familiarization session, the inclusion of a control group, and the use of a more sensitive and

validated methodology for motor unit tracking provides robust support that the observed changes reflect genuine neural adaptations to training, rather than task learning.

3. The average increase in the strength of the dorsiflexor muscles was 17.6% after the 4-week intervention, which is somewhat less than the 36% increase reported by Kamen & Knight (2004) for the knee extensors after a 6-week intervention. The current study reported an average increase in discharge rate of 11.3% during the plateau phase of the submaximal trapezoidal contractions, although the target force is not specified. In contrast, Kamen & Knight (2004) found no statistically significant change in discharge rate relative to the second baseline test during MVCs performed after the 6-week intervention. It is also puzzling that the current study reported a significant correlation ($r=0.54$) between the changes in MVF and discharge rate during the submaximal trapezoidal contractions. The differences between these two studies needs to be reconciled.

REPLY: We thank the reviewer for raising this point and appreciate the opportunity to clarify the differences with the study by Kamen & Knight (2004). While both studies investigated motor unit adaptations in older adults after short-term strength training, there are important differences in study design, recording techniques, tested muscle group, training protocols and interpretation that help contextualize the findings.

Firstly, the strength gains observed in our study (+17.6%) are smaller than those reported by Kamen & Knight (36%) after the 6-week dynamic intervention. As discussed in the manuscript (LL 731-738), this discrepancy likely reflects several factors: 1) the shorter duration of our intervention (4 vs. 6 weeks), 2) the nature of the training (isometric vs. dynamic), and 3) the muscle group involved (ankle dorsiflexors vs. knee extensors), the latter of which may have a greater potential for strength gains due to their larger size and primary role in weight-bearing activities. Supporting this, a more recent study by Orsatto et al. (2023), reported larger MVF increases (+25%) than those observed in our study after 6 weeks of power-oriented training. Conversely, Christie and Kamen (2010), who adopted a comparable isometric training protocol targeting ankle dorsiflexors for 2 weeks, reported a similar improvement in strength (+13%), aligning more closely with our findings.

Secondly, regarding discharge rate adaptations, Kamen & Knight (2004) reported a 49% increase in peak discharge rate between the first baseline and post-test during MVCs in older adults, but no significant change between the second baseline and post-test (i.e. training period). The authors attributed this to early neural adaptations or task learning. Importantly, no changes in discharge rate were reported at submaximal intensities (10% and 50% MVC) across any time point. Thus, discharge rate adaptations in their study appear to be task-dependent and restricted to maximal efforts.

In contrast, we observed an average 10.2% increase in discharge rate during submaximal contractions (specifically, +8.2% in the recruitment and +11.3% in the plateau phase). These differences likely reflect not only task specificity and the tested muscle group, but also key methodological differences. The use of HDsEMG decomposition combined with longitudinal motor unit tracking allowed us to identify and track a larger and more representative pool of motor units across sessions. This approach provides greater spatial resolution and sensitivity to detect discharge rate adaptations compared to traditional intramuscular EMG, which samples fewer motor units and does not allow the tracking of the same motor units over time.

Furthermore, although we did not analyze motor unit discharge rate during MVCs (i.e. peak discharge rate), this choice was motivated by the well-documented limitations in decomposing HDsEMG signals at very high contraction intensities (> 70-80% MVF) due to increased superimposition of independent sources (Del Vecchio et al. J Electromyogr kinesiol, 2020; Holobar & Farina, Physiol Meas, 2014). By contrast, submaximal contractions are known to elicit more stable and physiologically reliable discharge patterns and are less sensitive to inter-trial variability,

making them particularly suited to detect subtle adaptations in motor unit discharge behavior over time (Martinez-valdes et al., MSSE, 2018; Lecce et al., J Physiol, 2025; Valli et al., J Sport Health Sci, 2024).

Lastly, although the correlation between changes in MVF and discharge rate during submaximal contractions might seem unexpected, it is physiologically plausible. Discharge behavior at submaximal force levels reflects underlying motoneuron excitability and the strength of descending drive, both of which contribute to improvements in voluntary activation, a key determinant of strength gains in older adults (Rozand et al., 2020). Importantly, similar associations between submaximal motor unit discharge behavior and strength gains have been recently reported in young adults and even in untrained limbs, supporting the interpretation that such relationships primarily reflect neural adaptations rather than muscle-specific or task-specific changes (Lecce et al., J Physiol, 2025). Furthermore, our analysis relied on repeated-measure correlations based on longitudinal tracking of the same motor units, strengthening the inference that neural adaptations contributed to the improved force output, even when assessed under submaximal conditions.

As stated in the discussion (LL. 679-686), our results extend previous work by showing that motoneuron output can be modulated not only during maximal efforts but also at submaximal levels of activation. We believe that the differences between our findings and those of Kamen & Knight (2004) reflect the use of complementary but distinct methodologies, and that our study expands prior work by using a higher resolution motor unit recording technique and longitudinal tracking to capture training-induced changes in older adults at the individual motor unit level.

4. Were the estimates of persistent inward currents correlated with the increase in MVF?

REPLY: We thank the reviewer for this question. Changes in estimate of PICs (ΔF) were not directly correlated with changes in MVF. However, ΔF was significantly associated with training-induced changes in motor unit discharge rate, particularly during the plateau phase (Figure 5C), which in turn correlated with gains in MVF. This pattern is consistent with the established physiological role of PICs in amplifying and sustaining motoneuron firing rather than directly determining force output per se. Accordingly, increases in increased intrinsic excitability are expected to contribute to strength gains indirectly, through modulation of discharge behavior. Importantly, the presence or absence of a direct association between ΔF and force outcomes may depend on several factors (Mesquita et al., JNeurophysiol, 2024), including the muscle examined, its functional role, and the specific nature of the training stimulus. Previous studies reporting direct associations have indeed investigated different muscles (e.g. biceps brachii in Lecce et al., JPhysiol, 2025a, plantar flexors in Orssatto et al., JNeurophysiol, 2023) and employed distinct training paradigms (eccentric or power-oriented RT), which are known to differentially affect PIC-related mechanisms (Lecce et al., JPhysiol, 2025b).

In line with recent works (Mesquita et al., J neurophys, 2024; Lecce et al., EJAP, 2025), ΔF should therefore be interpreted primarily as an index of self-sustained firing capacity rather than a direct measure of motoneuron gain. We have now clarified this point in the Discussion (LL . 696-708).

5. If it is possible, develop a multiple-regression model to identify the outcome variables that can explain the increase in MVF.

REPLY: We thank the reviewer for this suggestion. While we agree that a multivariate approach can be more informative in some contexts, we did not perform a multiple regression analysis in the present study for both statistical and conceptual reasons. The candidate predictors of MVF changes (i.e. ΔF and motor unit discharge rate) were significantly intercorrelated, reflecting a physiological dependency whereby PIC-related changes influence force output indirectly through modulation of discharge behavior. Including such collinear predictors in a multiple regression model would violate model assumptions and complicate interpretation. In addition, the relatively small sample size of the intervention group ($n = 13$) would limit the robustness of multivariate

models. Instead, we adopted a *rmcorrelation* approach to test physiologically motivated associations, which better reflects the proposed mechanistic pathway linking intrinsic excitability, motor unit firing behavior and muscle strength gains. We hope we have clarified our perspective.

Minor Comments

70. State the test muscle.

REPLY: We have stated the tested muscle in the abstract as requested.

199. Define instantaneous.

REPLY: We have clarified the sentence in the revised manuscript to specify that “instantaneous” refers to the peak force value reached at any single time point during the maximal voluntary contractions.

470. All three target forces?

REPLY: We confirm that the average discharge rates shown in panels A (Intervention) and B (controls) were computed from motor units tracked over time across all three contraction intensities (35%, 50%, and 70% MVF). We have now specified this aspect in the Figure 2 legend.

574. This comparison requires evidence that the only difference between the two groups was chronological age, as implied.

REPLY: We thank the reviewer for reiterating this important point. As noted in our response to Major comment 1, we conducted a K-means clustering analysis based on baseline MVF, which did not yield significant association with age group. These findings are now clearly reported in the Results and referenced in the Discussion, to ensure age-related comparisons are appropriately contextualized.

631. Define "enhanced discharge behaviour".

REPLY: We have replaced “enhanced discharge behavior” with the more specific “increased discharge rates during submaximal contractions” in the revised manuscript, to better reflect the observed motor unit adaptations.

637. This sentence is misleading; report the changes after the two baseline sessions.

REPLY: We thank the reviewer for this comment and agree that the original sentence may have been misleading. As correctly pointed out, the 49% increase in maximal discharge rate reported by Kamen & Knight (2004) - as stated in their abstract and shown in Figure 3 - was calculated from the first baseline (~17 pps) to the post-training test (~25 pps), whereas no significant change was observed between the second baseline and post-training test. We have clarified this point in the revised manuscript to ensure accurate representation of their findings.

Referee #2:

The present study investigated motor unit adaptations underlying strength gains following a four-week isometric strength training program in older adults. The authors completed excellent works, and this knowledge would contribute to various research area and exercise or rehabilitation-associated fields. However, some part needs more detailed physiological considerations. This version just reported the results of older adults and comparison with the results of young adults. More discussion would help to make this article higher scientific value.

REPLY: We sincerely thank the reviewer for the positive evaluation of our work and for the highly constructive comments. We have revised the manuscript in line with the reviewer's suggestions. In particular, we have expanded the physiological framework used to interpret the motor unit adaptations in older adults, clarified age-specific mechanisms underlying the training responses and refined the comparison with young adults. We believe that these additions have substantially increased the scientific value and physiological depth of the manuscript.

Abstract

L78-80 - This part would induce misunderstanding that the present study used both young and older groups. The authors must not describe the results of young adults in the abstract.

REPLY: We thank the reviewer for this comment. We agree that the original wording could be misinterpreted as implying the inclusion of both young and older adults in the present study. We have therefore revised the Abstract to remove any description of results from young adults and to focus exclusively on the experimental data obtained in older participants involved in this study.

L108-115 - Some previous studies would reported characteristic motor unit activation pattern during acute motor tasks (ex. Nishikawa et al. Exp Gerontol 181: 112283, 2023). These findings help the authors to assume or hypothesize the characteristic age-related neural adaptation following chronic exercise intervention.

REPLY: We thank the reviewer for this suggestion. We agree that including recent evidence on acute neural responses may help better contextualize our hypothesis. Accordingly, we have incorporated a sentence in the Introduction (LL 138-141) referencing Nishikawa et al. (2023), who observed differential changes in MU discharge rate following a single bout of resistance exercise in older vs. young adults.

L123 - If the authors include the intervention studies with nutritional supplementations, there are various reports that the motor unit adaptations in older adults following resistance training (Nishikawa et a. Eur J Nutr 64: 117, 2025, Watanabe et al. Journal of Gerontology Series A: Biological Sciences 75(5): 867-874, 2020). "Scarce" would be insufficient to review the related research.

REPLY: We thank the reviewer for this suggestion and we agree that the inclusion of resistance training studies combined with nutritional supplementation provides a broader context for evaluating motor unit adaptations in older adults. Accordingly, we have revised the sentence to acknowledge this body of literature and replaced "remain scarce" with a more accurate statement in the revised manuscript (LL 123-125). We have also added citations to the studies by Watanabe et al. (2020) and Nishikawa et al. (2025) to reflect these contributions.

L145 - The descriptions of characteristic neural excitability in older adults were not fully reviewed in the Introduction. Also, the reason that the authors hypothesize similar pattern with young adults is unclear. Please describe physiological explanation for this hypothesis.

REPLY: We thank the reviewer for this comment. In the revised manuscript, we have expanded the Introduction to provide a more detailed description of age-related changes in motoneuron

excitability and motor unit behavior. This includes additional discussion of impairments in descending drive, reductions in intrinsic excitability, and motor unit remodeling associated with ageing.

Furthermore, we have clarified the rationale for our hypothesis by including a physiological justification (LL 137-145): although aged motoneurons are slower and less excitable, previous studies have demonstrated that they retain a capacity for neural plasticity following resistance training. This is supported by evidence of increased motor unit discharge rate, reduced afterhyperpolarization duration, and enhanced estimates of intrinsic excitability (e.g. PICs). Accordingly, we hypothesized that older adults would exhibit a qualitatively similar pattern of neural adaptation to that previously observed in young adults when exposed to the same training protocol.

L334 - How did the authors control the effects of antagonistic muscles? The older adults generally show greater antagonistic muscle activities.

REPLY: We thank the reviewer for raising this important point. While antagonist muscle coactivation was not directly assessed in the present study, we now acknowledge this limitation in the Discussion as a potential factor influencing net force output. However, current evidence suggests that changes in antagonist coactivation seem to contribute minimally to training-induced strength gains in older adults (Walker et al., *Exper Gerontol*, 2021). Reductions in coactivation are not consistently observed after resistance training and are unlikely to be a major determinant of increased voluntary force output, particularly in short-term interventions (< 4-6 weeks). Moreover, the functional role of antagonist activity in joint stabilization (Baratta et al., *Am J Sports Med*, 1988), which is generally preserved in older individuals, may limit its modulation in response to training. We have now integrated this consideration in the revised manuscript (LL 738-745).

L402-412 - Figure 1 is repeatedly used in the Methods and Results. It would be better to use once in Methods or Results.

REPLY: We thank the reviewer for this suggestion. In the revised manuscript, we now refer to Figure 1 only once, in the Methods section, where the experimental setup, decomposition, and motor unit tracking procedures are fully described. Accordingly, the corresponding paragraph in the Results section has been shortened to avoid redundancy.

Figure 2 A and B - Individual plots are difficult to identify. How many plots did include in this figure? Smaller plot markers might help to show all plots.

REPLY: We thank the reviewer for this suggestion. Figure 2A and 2B include all tracked motor units across the three contraction intensities (35%, 50%, and 70% MVF), grouped by participant. In total, 242 motor units were tracked in the intervention group and 161 in the control group, each color-coded by participant. To improve clarity and allow better visual separation of individual motor units, we have reduced the marker size in all panels. Additionally, we have also specified in the legend the total number of motor units represented. We believe this improves the visual distinction of individual data points.

Figures 4 and 5BC - Means of two plots within a participant were not described, i.e., Before and After. We can assume that left lower point would be Before and another could be After. However, it would be better to show it.

REPLY: We thank the reviewer for this observation. To improve clarity, we have revised the figure legends of both Figure 4A-B and Figure 5B-C to explicitly indicate that the two data points per participant represent pre- and post-training values, and that the paired values are connected with a line (with the lower dot corresponding to pre-training and the upper to post-training). These additions clarify the within-subject structure of the plots without modifying the visual layout. The updated legends are now included in the revised manuscript.

L574 - Brief descriptions of detailed data set should be stated in this part, i.e., number of tracked MUs for young adults.

REPLY: We thank the reviewer for this suggestion. We have now included a brief description of the datasets used in the comparative analysis, specifying the number of participants and the number of longitudinally tracked motor units for both young and older adults (LL 596-598).

L635-716 - As pointed in hypothesis, the authors didn't consider the age-related physiological responses or physiological conditions. This part of Discussion would be also able to be written from characteristics of older adults. Description that the results are similar with previous studies or the studies in young adults could be important, but comparisons with young adults are well documented in next session. Therefore, this part can be focused on the discussion for age-related physiological responses. Since baseline of motor unit activation properties should be affected in older adults, the authors must start from the summary of motor unit firing properties in older adults.

REPLY: We have revised this section of the Discussion to explicitly start from the baseline physiological characteristics of motor unit behavior in older adults and to interpret the observed training-induced adaptations within this age-specific context. Moreover, direct comparisons with young adults have been moved to the subsequent paragraph.

L724-735 - Although similar responses were detected between young and older adults, detailed responses were not same. These differences can be detected by motor unit decomposition with HD-SEMG and strength point of this study. Thus, the authors must not conclude that older adults showed similar adaptations with young adults. Please add the discussion for the differences in recruitment threshold and/or a significant increase in discharge rate at recruitment.

REPLY: We thank the reviewer for this observation. We have revised this section of the discussion accordingly (LL. 762-776), to avoid concluding that young and older adults showed similar motor unit adaptations. This paragraph has been modified to more explicitly highlight age-specific peculiarities of training-induced motor unit adaptations. Specifically, we now discuss the attenuated modulation of recruitment threshold in older adults and the significant increase in discharge rate at motor unit recruitment observed only in this group, suggesting age-specific strategies to enhance motor unit output. In line with these modifications, we also updated the Abstract, Key Points and Conclusions to ensure consistent terminology and to avoid implying identical motor unit adaptations between young and older adults.

L738-745 - More physiological discussion should be needed for this part. Why were this difference observed in plateau phase? As pointed above, the authors can cite some research that showed difference in acute response of neuromuscular activation between young and older adults.

REPLY: Thank you for the suggestion. We have integrated the Discussion to provide a more detailed physiological interpretation of the age-related difference observed during the constant-force (plateau) phase (LL. 779-790). Specifically, we now discuss why the training-induced increase in discharge rate during sustained force production was attenuated in older adults, linking this finding to age-related constraints in sustaining high-frequency motor unit firing. In line with the reviewer's suggestion, we have also cited evidence from acute studies (e.g. Nishikawa et al., 2023) showing age-related differences in neuromuscular responses following resistance exercise, which may help explain the reduced magnitude of adaptation observed in older adults during the plateau phase.

Dear Professor Del Vecchio,

Re: JP-RP-2026-290541R1 "**Ageing Does Not Impair Motor Neuron Adaptations: Comparable Motor Unit Responses to Strength Training in Young and Older Adults**" by Andrea Casolo, Stefanie Del Vecchio, Benjamin Goodlich, Bastian Schrader, Stefano Nuccio, Edoardo Lecce, Ilenia Bazzucchi, Luca Angius, Francesco Felici, Joachim Schrader, Dario Farina, and Alessandro Del Vecchio

We are pleased to tell you that your paper has been accepted for publication in The Journal of Physiology.

- The contact information for the person responsible for 'Research Governance' at your institution needs to be provided. This includes their name and an institutional email address. Please ensure the contact is not an author on this paper and provide an alternate contact if necessary, or confirm in the submission form that the author whose email was provided has sole responsibility for research governance. This is the person who is responsible for regulations, principles and standards of good practice in research carried out at the institution, for instance the ethical treatment of animals, the keeping of proper experimental records or the reporting of results.

Yours sincerely,

Karyn Hamilton
Senior Editor
The Journal of Physiology

IMPORTANT POINTS TO NOTE FOLLOWING ACCEPTANCE OF YOUR PAPER:

- **IMPORTANT NOTICE ABOUT OPEN ACCESS:** To assist authors whose funding agencies mandate immediate public access to published research findings, The Journal of Physiology allows authors to pay an Open Access (OA) fee to have their papers made freely available immediately on publication.

- You can help your research get the attention it deserves! Check out Wiley's free Promotion Guide for best-practice recommendations for promoting your work at: www.wileyauthors.com/eoo/guide. You can learn more about Wiley Editing Services which offers professional video, design, and writing services to create shareable video abstracts, infographics, conference posters, lay summaries, and research news stories for your research at: www.wileyauthors.com/eoo/promotion.

- If you would like to receive our 'Research Roundup', a monthly newsletter highlighting the cutting-edge research published in The Physiological Society's family of journals (The Journal of Physiology, Experimental Physiology, Physiological Reports, The Journal of Nutritional Physiology and The Journal of Precision Medicine: Health and Disease), please click this link, fill in your name and email address and select 'Research Roundup': <https://www.physoc.org/journals-and-media/membernews>

EDITOR COMMENTS

Reviewing Editor:

Comments to the Author (Required):

Thank you for your thorough response to the critiques. Both reviewers are satisfied and speak highly of your work. Congratulations, and thank you for submitting your best work to The Journal of Physiology.

Senior Editor:

Comments to the Author:

Thank you for submitting your revised manuscript for continued consideration by The Journal of Physiology. The Referees were both complimentary about the improvements resulting from the revisions. We are pleased to accept it for publication in The Journal of Physiology. Thank you for your interest in The Journal and Congratulations!

REFEREE COMMENTS

Referee #1:

The responses have addressed my concerns.

Referee #2:

The authors completely revised the paper following my previous comments.